# QiMeng-LibBench: Benchmarking LLM Agents for Library-Scale Cross-Architecture Migration

**Weijia Li** [1 2] **Ke Gao** [1 2] **Jiajie Li** [1 2] **Han Sun** [1 2] **Yuhe Ding** [1 2] **Yongdong Mai** [1 2] **Yiran Le** [1 2] **Yongjie Qian** [1 2]
**Zhibin Zhang** [1 2] **Xinyu Wang** [1 2] **Limin Cheng** [1 2] **Shouxu Kuang** [1 2] **Pengfei Chen** [1 2] **Ling Li** [1 2]

## Abstract

Cross-architecture migration of high-performance libraries dictates ecosystem readiness on emerging hardware. The challenge is twofold: disentangling library-scale dependencies and performance-critical kernels with ISA-specific SIMD intrinsics, often trading migration speed for peak performance. While LLM-based agents offer a promising approach, are confined to function-level tasks or scalar code, failing to assess agents' capabilities and limitations in realistic, library-scale migration. We present QM-LibBench, a benchmark for cross-architecture library-scale code migration, featuring 85 critical kernels from widely used libraries, including OpenCV, libjpeg, and NCNN. It supports comprehensive evaluations of compilability, correctness, and performance across major transitions: ARM→RISC-V, x86→ARM, and ARM→LoongArch. Evaluation of 12 SOTA agent-LLM combinations on QM-LibBench reveals that, due to the lack of library-level navigation and hardware-aware optimization, agents regress to superficial pattern matching, yielding only 20.88% correctness and 0.83 speedup for libjpeg. Motivated by these findings, we further propose FSCM, a multi-agent framework incorporating hardware-aware global reconfiguration and performance optimization. FSCM improves OpenCV correctness to 71%. The benchmark and code are available at https://github.com/WisdomJoy/QM-LibBench.

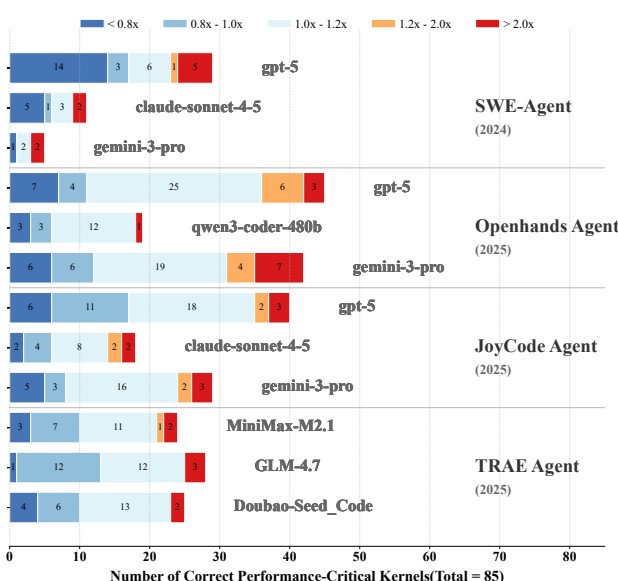

Figure 1. Overall Performance Comparison of Agents(SR@3).

[1]Intelligent Software Research Center, Institute of Software, CAS, Beijing, China [2]University of Chinese Academy of Sciences, Beijing, China. Correspondence to: Ling Li <liling@iscas.ac.cn>.

*Proceedings of the 43rd International Conference on Machine Learning*, Seoul, South Korea. PMLR 306, 2026. Copyright 2026 by the author(s).

## 1. Introduction

The growing demand for deploying compute-intensive tasks on heterogeneous platforms has accelerated the spread of multiple Instruction Set Architectures (ISAs), such as ARM and RISC-V. This trend necessitates the efficient migration of high-performance libraries like OpenCV and ncnn to various platforms. However, library-scale migration far exceeds simple recompilation. It demands consistent modifications across million-line codebases with intricate dependencies, as well as the rewriting of architecture-specific SIMD intrinsics in performance-critical kernels.

Existing approaches fall short of library-scale, cross-architecture migration. Rule-based tools such as neon2rvv(Hau, 2025) depend on rigid one-to-one intrinsic mappings and lack global semantic understanding, yielding inefficient or even incorrect code. LLM agents have shown promise for automating software engineering tasks, but the evaluation benchmarks either prioritize functional correctness or focus on localized transformations. For in-

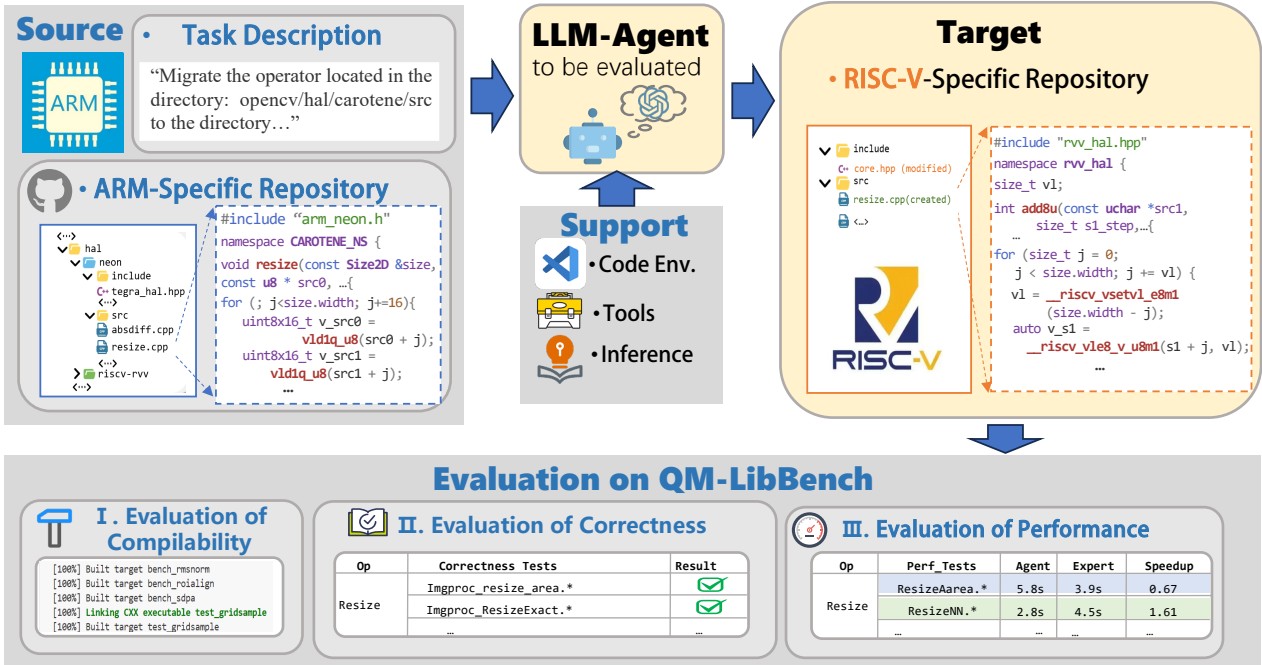

*Figure 2.* QM-LibBench migration and evaluation pipeline. An LLM agent performs library-level cross-ISA migration from a source library to a target architecture. The generated code is evaluated end-to-end on compilability, functional correctness, and performance, with SIMD kernel translation illustrating the shift from fixed-width to vector-length-agnostic execution.

stance, SWE-Bench targets scalar bug-fixing and ignores architecture-specific vectorization, while SIMDBench is intrinsic-centric but omits project-level dependencies. These gaps motivate a dedicated benchmark that evaluates agents in realistic migration settings.

To bridge this gap, we introduce QM-LibBench, a new benchmark designed to evaluate LLM agents on realistic, cross-architecture migration tasks. and provide comprehensive evidence of current agents' capabilities and limitations. Our contributions are:

- **Benchmark Construction:** We release QM-LibBench, the first library-scale benchmark for cross-architecture code migration. It comprises 85 performance-critical kernels mined from industrial libraries (OpenCV, libjpeg, ncnn) across RISC-V, ARM, and LoongArch, evaluating agents on correctness and performance within a full library context.

- **Comprehensive Evaluation:** We evaluate 12 agent–LLM combinations, revealing that current models rely heavily on superficial pattern matching. They struggle with library-level dependencies and hardware specifics, resulting in low correctness (20.8%) and suboptimal performance (0.83× performance for libjpeg).

- **Methodological Validation:** To address these limita-

tions, we propose FSCM (Faster and Smarter Code Migration), a hardware-aware global reconfiguration and optimization framework. FSCM significantly boosts OpenCV correctness to 71%, confirming that solving QM-LibBench requires moving beyond translation to explicit architectural optimization.

## 2. Background

### 2.1. The Challenges of Migrating SIMD Intrinsics

In data-level parallelism, existing approaches such as compiler auto-vectorization and handwritten assembly suffer from either conservativeness or composability. SIMD intrinsics therefore emerge as a practical trade-off, providing explicit control over vector operations while preserving high-level language structure. As a result, they have become a widely adopted technique for performance optimization in high-performance libraries. However, direct translation across architectures for SIMD-intrinsic codebases is severely constrained by the combinatorial differences in vector register width, semantic behavior, and hardware constraints, leading to extremely low success rates. Consequently, effective code migration requires a deep structural understanding of the original implementation, including data layout transformations and algorithmic redesign.

*Table 1.* Comparison of QM-LibBench with related benchmarks

| Benchmark | Problem Scale | | | Evaluation Metrics | | Task Source | |
| --- | --- | --- | --- | --- | --- | --- | --- |
| | Granularity | Input(LoC) | Output(LoC) | Correctness | Performance | Cross-Arch | Real-World |
| Humaneval | Func-level | NL description | 6 | ✔ | ✗ | ✗ | ✗ |
| ClassEval | Class-level | 51 | 35 | ✔ | ✗ | ✗ | ✗ |
| SWE-bench | Repo-level | 438K | 33 | ✔ | ✗ | ✗ | ✔ |
| KernelBench | Repo-level | 49 | 62 | ✔ | ✔ | ✗ | ✗ |
| Swe-Perf | Repo-level | 170K | 131 | ✔ | ✔ | ✗ | ✔ |
| SIMDBench | Func-level | NL description | 10.7 | ✔ | ✔ | ✔ | ✗ |
| VecIntrinBench | Func-level | 140 | 90 | ✔ | ✔ | ✔ | ✔ |
| **QM-LibBench** | **Repo-level** | **717K** | **991** | ✔ | ✔ | ✔ | ✔ |

## 2.2. Migration Target Architectures

We consider four representative architectures for migration and their associated SIMD instruction sets, covering a broad range of cross-architecture migration scenarios.

**x86 (AVX).** As the most mature and widely deployed general-purpose architecture, x86 with AVX often serves as the source platform for SIMD-based implementations.

**ARM (NEON).** ARM dominates mobile and embedded platforms, with NEON providing a distinct SIMD model tailored to low-power execution.

**RISC-V (RVV).** Although RISC-V is rapidly evolving, it remains relatively immature. RVV adopts a vector-length-agnostic execution model.

**LoongArch (LASX).** LASX is a SIMD extension for LoongArch with a relatively underdeveloped software ecosystem.

These instruction sets differ fundamentally in vector abstractions and ecosystem maturity. AVX is typically fixed-width. NEON's vector-splitting and rearrangement differ from those of others. RVV introduces dynamic vector lengths that invalidate many fixed-width assumptions, while LASX lacks sufficient reference implementations. These discrepancies make direct translation unreliable and require a structural understanding during cross-architecture migration.

## 2.3. Complexity of Library Migration

High-performance libraries such as OpenCV and ncnn typically combine generic scalar code with multiple architecture-specific implementations for the same operator. At runtime, different implementations are selected depending on the underlying hardware. During migration, the agent should correctly dispatch the operators to the target-specific implementations rather than follow the legacy behaviors of the source code. However, operators are often tightly coupled with the source architecture's data layout and memory alignment. As a result, a successful migration requires an understanding of the algorithmic semantics and the library structure.

## 3. QM-LibBench

QM-LibBench evaluates library-level migration by focusing on performance-critical kernels—core operators in high-performance libraries (e.g., codec primitives, image processing kernels, and neural network operators). The migration process is shown in Figure 2.

## 3.1. Task Formulation

Formally, a migration task is defined as a tuple $\mathcal{T} = (I, \mathcal{R}_{src}, \mathcal{E})$, where $I$ denotes the natural language specification (target operator, path, and ISA), $\mathcal{R}_{src}$ denotes the source library (file structure and build configurations), and $\mathcal{E}$ is the execution environment of the agent. Given $T$, the agent's goal is to synthesize a target library $\mathcal{R}_{tgt}$ and ensure the new implementation is dependency-compliant. We evaluate $\mathcal{R}_{tgt}$ on three orthogonal dimensions: Compilability, Functional Correctness, and Performance.

## 3.2. Dataset Collection

While SIMD intrinsics are ubiquitous in modern industrial libraries for achieving vectorization acceleration, constructing a benchmark directly from these raw repositories is non-trivial due to the entangled nature of real-world software, including complex cross-file dependencies and diverse hardware abstraction layers. To ensure the dataset comprises high-quality and testable cases, we devise the following construction process, which involves three phases.

**Phase 1: Source Library Selection.** Diverse target libraries that are domain-representative are prioritized, including OpenCV for computer vision workloads, ncnn as a lightweight deep learning inference framework, and lib-jpeg as a widely used image compression library. These libraries feature real-world deployment and continuous efforts in manual SIMD-optimization, offering a rich testbed for cross-architecture migration tasks.

**Phase 2: Operator Selection and Validation.** Large-scale open-source libraries often suffer from *code rot* due to prolonged maintenance and distributed collaboration, contain-

*Table 2.* Composition of QM-LibBench

| Library | Source | Target | Scale | Count |
|---------|--------|-----------|-------|-------|
| OpenCV | ARM | RISC-V | | 21 |
| ncnn | x86 | ARM | | 17 |
| | ARM | RISC-V | | 17 |
| | ARM | LoongArch | | 17 |
| libjpeg | ARM | RISC-V | | 13 |

ing much irrelevant or deprecated code. Instead of file-by-file validation, we dynamically trace operator files by running the official test suites of each library, only keeping invoked and verified operators. This step ensures that the selected cases are both functional and actively maintained.

**Phase 3: Operator Filtering.** We filter the operators based on functional centrality in the library and the criticality of performance optimization. Operators that are most frequently invoked in workloads, such as `Resize` in OpenCV, are prioritized to ensure that the migration benchmarks reflect realistic usage patterns. Furthermore, operators for which SIMD vectorization provides substantial optimization are prioritized. Their implementations are inherently designed around SIMD intrinsics, with control flow and data layouts co-optimized for vector execution.

### 3.3. Test Construction

QM-LibBench establishes a 2-phase evaluation for each task, encompassing both functional correctness and performance.

**Correctness.** The objective of this phase is to validate the functional equivalence between the agent-generated SIMD code and the ground-truth scalar implementation. For OpenCV and ncnn, we leveraged their established testing suites, manually extracting test cases corresponding to each selected operator. For libjpeg, which lacks operator-level tests, we developed a test suite with Google Test.

**Performance.** For OpenCV, which provides a mature performance benchmark, we similarly filtered and reused the official benchmarks relevant to our dataset. For ncnn and libjpeg, where performance benchmarks were insufficient, we constructed custom benchmarks using Google Benchmark, covering a wide range of input dimensions and typical parameters. This ensures that the performance metrics reflect the operator's behavior under realistic workloads rather than isolated edge cases.

### 3.4. Evaluation Metrics

To rigorously evaluate effectiveness in cross-architecture code migration, we consider three key metrics:

**Correctness.** We use SR@$k$(Success Rate @ turn $k$) as our evaluation metric, where an agent is allowed up to $k$ independent generation attempts per task. SR@$k$ better reflects an agent's practical usability in realistic development scenarios.

**Performance.** We use the speedup ratio (SR) to evaluate the relative execution performance of the migrated implementation compared to a vanilla baseline. We compute SR as:

$$\text{SR} = T_{\text{original}}/T_{\text{migrate}} \tag{1}$$

where $T_{\text{original}}$ is the execution time of the original code, and $T_{\text{migrate}}$ is that of the migrated implementation. Both $T_{\text{original}}$ and $T_{\text{migrate}}$ are measured on the same target hardware. $T_{\text{original}}$ refers to the execution time of the existing target-architecture implementation in the original library.

**Joint Performance–Correctness Metric.** We adopt fast$_p$ (Ouyang et al., 2025) as the fraction of migration tasks whose outputs are functionally correct and achieve a speedup of at least $p$ over a baseline:

$$\text{fast}_p = \frac{1}{N} \sum_{i=1}^{N} \mathbb{I}(\text{Success}_i \wedge (\text{SR}_i \geq p)). \tag{2}$$

Here, $\text{Success}_i$ indicates successful compilation and correctness of the $i$-th case on the target architecture. $\mathbb{I}$ is the indicator function that equals 1 when the constraint is satisfied and 0 otherwise. Larger $p$ values reflect stronger performance optimization.

## 4. Experimental Setup

**Hardware Platforms.** We conduct our evaluation on three platforms: Cortex-A76 (ARM, 2018), SpacemiT-X60 (SpacemiT, 2024), and LoongSon 3A6000 (LoongSon, 2024), covering established and emerging architectures, including ARM, RISC-V, and LoongArch.

**LLM Agents.** We base our agent selection on SWE-Bench, which has become a de facto standard for evaluating library-level code repair and automated software engineering systems. According to publicly reported SWE-Bench leaderboard, we select four widely used and empirically strong code agents—TRAE IDE(ByteDance, 2025a), SWE-Agent(Yang et al., 2024), JoyCode IDE(JD, 2025), and OpenHands(Wang et al., 2025c)—as representative state-of-the-art systems.

**Model and Constraints** The underlying LLMs for each agent are chosen based on a combination of empirical performance and platform constraints. (1) For SWE-Agent and JoyCode, we employ the current leading proprietary LLMs: GPT-5(OpenAI, 2025), Gemini-3-Pro-Preview(DeepMind, 2025), and Claude-Sonnet-4.5(Anthropic, 2025). (2) For

*Table 3.* Correctness Results Across Libraries and Architectures Migration Settings. Shaded columns denote overall averages. **Bold** indicates the best-performing model under each agent, and values mark the best result among all agents.

| Models | Cross Libraries | | | | | | | | Corss Architectures | | | | | |
| | OpenCV | | ncnn | | libjpeg | | overall | | x86→ARM | | ARM→RISC-V | | ARM→LoongArch | |
| | SR@1 | SR@3 | SR@1 | SR@3 | SR@1 | SR@3 | SR@1 | SR@3 | SR@1 | SR@3 | SR@1 | SR@3 | SR@1 | SR@3 |
| *SWE-Agent (Yang et al., 2024)* | | | | | | | | | | | | | | |
| + claude-sonnet-4-5 | 0.00 | 9.52 | 7.84 | 19.61 | 0.00 | 0.00 | 3.53 | 12.94 | 11.76 | 35.29 | 0.00 | 11.76 | 5.88 | 5.88 |
| + gemini-3-pro-preview | 0.00 | 4.76 | 5.88 | 11.76 | 0.00 | 0.00 | 3.53 | 5.88 | 5.88 | 11.76 | 5.88 | 5.88 | 5.88 | 5.88 |
| + GPT-5 | **9.52** | **23.80** | **35.29** | **50.98** | 0.00 | 0.00 | **22.35** | **34.12** | 52.94 | 58.82 | 11.76 | 35.29 | 35.29 | 47.06 |
| *JoyCode (JD, 2025)* | | | | | | | | | | | | | | |
| + claude-sonnet-4-5 | 9.52 | 19.05 | 9.80 | 25.49 | 0.00 | 7.69 | 8.24 | 21.18 | 5.88 | 23.53 | 23.53 | 35.29 | 0.00 | 17.65 |
| + gemini-3-pro-preview | **19.05** | 19.05 | 25.49 | 41.18 | 0.00 | **30.76** | 20.00 | 35.29 | 41.18 | 58.82 | 5.88 | 23.53 | **29.41** | 41.18 |
| + GPT-5 | 14.29 | **28.57** | 37.25 | 58.82 | 0.00 | 30.76 | 25.88 | 47.06 | 58.82 | 64.71 | 29.41 | 52.94 | 23.53 | **52.94** |
| *Openhands (Wang et al., 2025c)* | | | | | | | | | | | | | | |
| + qwen3-coder-480b | 19.05 | 23.81 | 17.65 | 23.53 | 7.69 | 15.38 | 17.65 | 23.53 | 29.41 | 35.29 | 11.76 | 23.53 | 11.76 | 11.76 |
| + gemini-3-pro-preview | 23.81 | **42.86** | 43.14 | 54.90 | 30.77 | 38.46 | 36.47 | 49.41 | 58.82 | 64.71 | 35.29 | 52.94 | 35.29 | 47.06 |
| + GPT-5 | 28.57 | 38.10 | 41.18 | **60.78** | 38.46 | 46.15 | **37.65** | 52.94 | 41.18 | 52.94 | 23.53 | 64.71 | 58.82 | 64.71 |
| *TRAE (ByteDance, 2025a)* | | | | | | | | | | | | | | |
| + Doubao-Seed-Code | 19.05 | 23.81 | 17.65 | 31.37 | **30.77** | **30.77** | 23.53 | 35.29 | 11.76 | 35.29 | **23.53** | **41.18** | 17.65 | 17.65 |
| + GLM-4.7 | **23.81** | **42.86** | 23.53 | 33.33 | 15.38 | 15.38 | 22.35 | 36.47 | **35.29** | **47.06** | **23.53** | 35.29 | 11.76 | 17.65 |
| + MiniMax-M2.1 | 9.52 | 9.52 | 19.61 | 39.22 | 0.00 | 15.38 | **27.06** | **47.06** | 23.53 | 35.29 | **23.53** | **41.18** | 11.76 | **41.18** |
| **Average** | 14.68 | 23.81 | 23.37 | 36.60 | 10.26 | 19.23 | 20.88 | 33.92 | 31.37 | 43.63 | 18.14 | 35.29 | 20.59 | 30.88 |

OpenHands, we replaced Claude with Qwen3-Coder-480b-a35b-Instruct (Qwen, 2025) as preliminary tests indicated that the combination failed to transition from planning to code generation in our tasks. (3) For TRAE IDE, due to its integrated ecosystem constraints, we selected its supported LLMs: Doubao-seed-Code(ByteDance, 2025b), GLM-4.7(Zhipu, 2025), and MiniMax-M2.1(MiniMax, 2025). Notably, the combination of TRAE and Doubao-seed-Code currently holds the top position on the SWE-Bench leaderboard. We set a timeout of over 40 minutes to prevent the agent from continuing its generation. Detailed configurations are provided in Appendix B.

**Rule-based Baselines.** We do not include rule-based migration tools such as neon2rvv as primary baselines because they are not directly comparable to autonomous LLM agents in library-scale migration settings. Applying such tools requires substantial manual engineering effort, including modifying build systems, restructuring headers, adding platform detection, and defining architecture-specific macros. Moreover, we observed that neon2rvv consistently fails on QM-LibBench due to fundamental ISA mismatches between fixed-width NEON and vector-length-agnostic RVV. For example, RVV vector types lack compile-time fixed sizes, causing failures in structs/unions, while NEON-style function overloading based on vector width often becomes invalid after naive translation. These issues reflect inherent limitations of rule-based intrinsic mapping for realistic library-scale migration.

## 5. Evaluation

### 5.1. Correctness

We evaluate the migration correctness of 12 agent-LLM combinations on QM-LibBench. Table 3 reports SR@1 and SR@3 results across 3 libraries and 3 cross-architecture migration scenarios. The evaluation reveals a significant capability gap in existing agents when addressing library-scale SIMD migration. Even the strongest combination, OpenHands powered by GPT-5, achieves an overall SR@1 of only 37.65%.

**Analysis of Agents.** SWE-Agent collapses across nearly all metrics as it inherits from the SWE-Bench workflow (Jimenez et al., 2023), which is issue-level, failing to transfer to library-scale C/C++ migration. OpenHands achieves the highest overall performance (37.65% SR@1 with GPT-5). We attribute this to its architecture based on CodeAct (Wang et al., 2024), which integrates LLM reasoning with executable command-line tools for iterative debugging. Joy-Code and TRAE, as IDE-based agents, improve stability through persistent project environments and stronger file-navigation support. They avoid collapse and achieve competitive SR@3 scores. Yet all these agents are still far from practical real-world migration tasks.

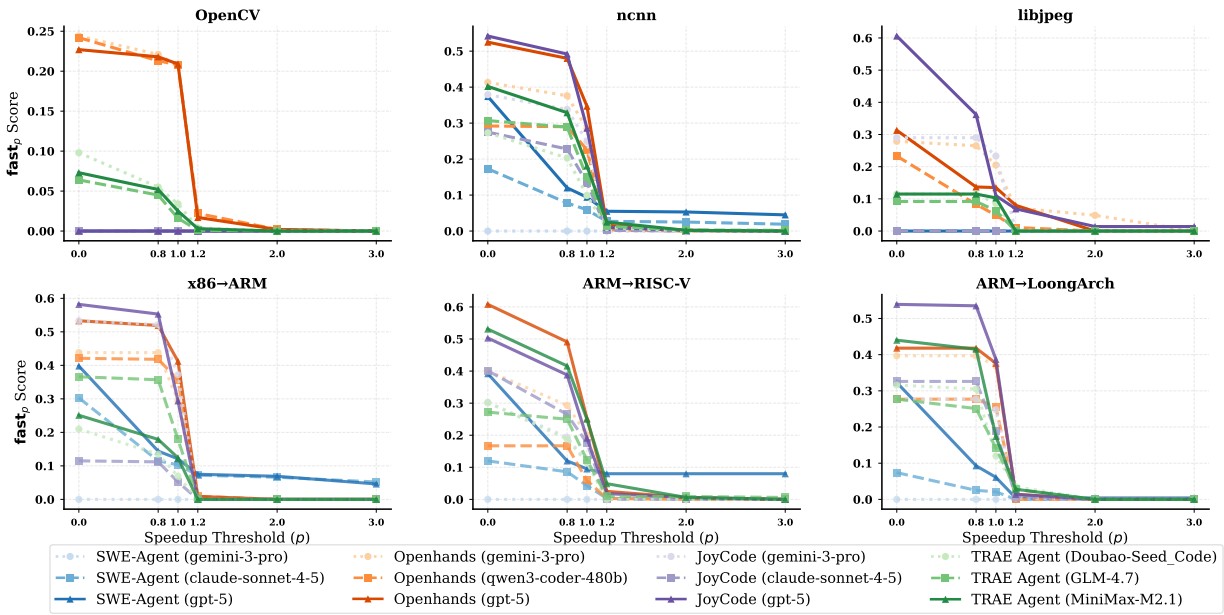

*Figure 3.* Performance–correctness trade-off measured by the $\text{fast}_p$ metric across libraries and migration settings. Each curve shows $\text{fast}_p$, the fraction of migration tasks whose outputs are functionally correct and achieve a speedup $\geq p$. Notably, $\text{fast}_0$ reduces to the proportion of functionally correct migrations.

**Analysis of LLMs.** The correctness of migration is strongly correlated with the foundation model's capability. The open-weights model Qwen3-coder-480b-a35b-Instruct lags behind proprietary models, achieving only 17.65% Overall SR@1 even within OpenHands. While Gemini-3-pro-preview performs competitively, it suffers from severe performance degradation when paired with less capable agents like SWE-Agent. Notably, GPT-5 consistently achieves the highest performance across all agents, yielding an average 10.39 improvement in SR@1 over the second-best model. This trend underscores that model capability is the dominant factor for migration quality.

**Analysis of Libraries.** Agents achieve the best results (Avg. 23.37%) in ncnn, likely attributed to its unified development toward a clear structure. OpenCV lags behind (Avg. 14.68%) due to its prolonged community maintenance and layered dependencies. libjpeg also proves challenging despite its smaller scale, as it heavily relies on precise bit-level manipulations and strict format constraints.

**Analysis of Architectures.** Migration between mainstream architectures (x86→ARM) yields the highest average accuracy (31.37%), attributed to the rich ecosystem. In contrast, targeting emerging architectures like RISC-V (18.14%) and LoongArch (20.59%) proves more difficult due to data scarcity.

## 5.2. Migration Performance

Beyond functional correctness, we evaluate the performance of migrated code using the $\text{fast}_p$ metric, which jointly measures correctness and speedup. Figure 3 reports $\text{fast}_p$ scores across varying speedup thresholds $p$ for different libraries and migration scenarios.

**Overall Performance Results.** As the speedup threshold $p$ increases from 0 to 1.0, $\text{fast}_p$ scores drop sharply across all agent-LLM combinations. At $p = 1.0$, merely requiring no performance degradation, most agents already exhibit near-zero scores. When $p > 1.2$, almost no agent produces code that achieves speedup. This indicates that existing agents prioritize functional correctness over performance, often generating inefficient or even slower implementations.

**Library-Specific Results.** On ncnn, agents achieve relatively higher $\text{fast}_p$ scores with a low $p$. On OpenCV and libjpeg the best-performing combinations barely exceed 0.25 and 0.30 at $p = 0$. Yet all agents rapidly collapse to 0 as $p$ increases. This suggests that agents are insufficient for real-world migration tasks where performance is critical, especially for complex libraries with intricate dependencies and file structures.

**Architecture-Specific Results.** ARM→RISC-V migrations suffer steeper degradation compared to the other two scenarios. This can be attributed to the paradigm mismatch (fixed-width to variable-width) and the immature ecosystem

of RISC-V RVV.

**Analysis of Agents and LLMs** OpenHands with GPT-5 and JoyCode with GPT-5 consistently achieve the highest $fast_p$ scores across most scenarios. In particular, JoyCode with GPT-5 leads on x86→ARM (0.55), while OpenHands with GPT-5 leads on ARM→RISC-V (0.60) and ARM→LoongArch (0.55). OpenHands with Gemini-3-pro also performs competitively, achieving around 0.40 across multiple scenarios. Notably, SWE-Agent with GPT-5 achieves the highest $fast_p$ on ncnn with a high $p$, despite its near-zero performance on other libraries and scenarios, which is still low. This can be attributed to ncnn's unified structure matching SWE-Agent's issue-level workflow.

*Table 4.* Cost Comparison (SR@1)

| Costs | JoyCode | Openhands | SWE-Agent |
|---|---|---|---|
| *Successful Migration* | | | |
| Time | 593 | 1764 | 585 |
| Money | 1.60 | 2.04 | 1.38 |
| API Calls | 20 | 81 | 36 |
| Input Tokens | 993K | 1850K | 1531K |
| Output Tokens | 36K | 121K | 2K |
| *Failed Migration* | | | |
| Time | 1156 | 1358 | 843 |
| Money | 2.14 | 2.19 | 1.71 |
| API Calls | 28 | 84 | 44 |
| Input Tokens | 1134K | 2110K | 1803K |
| Output Tokens | 71K | 237K | 13K |

### 5.3. Cost Analysis

Table 4 reports the costs on gpt-5 settings for the three agents with measurable cost statistics. The complete cost breakdown is provided in the appendix C. Although JoyCode exhibits slightly lower migration correctness than Openhands (Section 5.1), it delivers substantial gains in cost efficiency. For successful migrations, OpenHands requires up to 3× longer completion time and 4× more API calls, resulting in consistently higher monetary costs. This indicates that OpenHands trades efficiency for marginal correctness improvements, likely due to heavier planning, more frequent tool invocation, and longer self-reflection cycles.

Additionally, failures are more expensive across all agents. Unlike successful trajectories, which remain focused and goal-directed, failed ones tend to enter prolonged cycles of iterative debugging, speculative rewrites, and context expansion. Agents lack mechanisms to detect low-probability trajectories and terminate early, leading to disproportionate resource expenditure.

### 5.4. Failure Mode Analysis

Figure 4 illustrates the failure mode distribution across 966 samples. The primary obstacle involves the misuse of SIMD intrinsics, where *Undefined Identifiers* (21.74%), *Data Type Errors* (11.80%), and *Parameter Mismatches* (8.90%) collectively account for over 42% of invalid cases. This is largely driven by superficial pattern matching during migration, where agents generate "hallucinated" APIs that mimic legitimate naming patterns but remain undefined or incorrectly applied in the target SDK. *Functional Correctness Errors* (15.01%) follow, reflecting the inherent challenges of porting low-level vectorized logic such as data alignment and masking. Furthermore, a significant cluster of failures underscores the agents' insufficient grasp of complex library structures and planning: *Artifact Integrity & Configuration Defects* (12.94%) and *Process Execution Exceptions* (10.14%) reveal instabilities in multi-file delivery and long-horizon planning, while *Dependency & Build Errors* (10.04%) and *Interface Binding Failures* (4.55%) further highlight structural gaps where valid operator logic remains uncallable due to missing headers or misconfigured build dependencies.

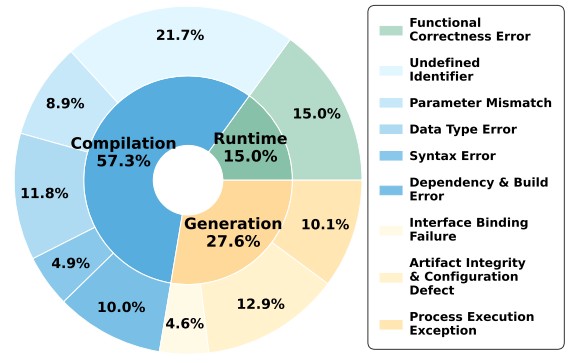

*Figure 4.* Overall Error Distribution.

### 5.5. Case Study

Figure 5 illustrates a representative failure mode observed in our evaluation. Red annotations highlight erroneous code segments, while green marks the correct counterparts. The agent tends to rely on superficial pattern matching during translation, leading to systematic errors. For instance, the code agent directly copies the source-specific parameter `const Size2D &size` without verifying its compatibility with the target library. Furthermore, the model exhibits false pattern association by generating `__riscv_vle16` (a 16-bit load) for 8-bit input data (u8*). This mismatch in-

```
#include "vtransform.hpp"
namespace CAROTENE_NS {                ARM code
void add(const Size2D &size, const u8 * src0, …{
   …
   for (; j < size.width; j += 16){
       …
       uint8x16_t v_src0 = vld1q_u8(src0 + j);
       uint8x16_t v_src1 = vld1q_u8(src1 + j);
       …
```

```
#include "rvv_hal.hpp"
namespace rvv_hal {                    Error code
int vl;
int add(const Size2D &size, const u8 *src0,…{
   …
   for (size_t j = 0; j < size.width; j += vl) {
       vl = setvl(size.width - j);
       …
       auto v_src1 = __riscv_vle16(s1 + j, vl);
       …
```

```
#include "rvv_hal.hpp"
namespace rvv_hal {                    Correct code
size_t vl;
int add8u(const uchar *src1, const uchar *src2,…{
   …
   for (size_t j = 0; j < size.width; j += vl) {
       vl = __riscv_vsetvl_e8m1(size.width - j);
       …
       auto v_s1 = __riscv_vle8_v_u8m1(s1 + j, vl);
       …
```

*Figure 5.* Error Code Example.

dicates that the agent is statistically imitating the numeric token `16` from the source loop rather than reasoning about the actual data types. Together with the hallucinated API call `setvl`, these errors demonstrate that the agent performs surface-level syntactic translation instead of the precise, library-aware migration required for systems-level code.

## 6. FSCM

To investigate whether surface-level pattern matching can be alleviated, we propose FSCM (Faster and Smarter Code Migration), a code migration framework based on core function mining and recursive code generation. FSCM refactors the source code and dependency relationships, enabling LLM agents to perform dependency-aware functionally equivalent migration and architecture-aware code optimization.

The overall workflow of FSCM is illustrated in Appendix E. FSCM consists of 4 major phases. **(1)** The *global parsing and core function mining* phase extracts core functionalities of the source code and restructures the repository into a Global Functional Dependency Graph (GFDG). This graph stores the semantic information of each functional module and its architecture-independent optimization strategies. **(2)** Guided by the GFDG, the *dependency-aware recursive code generation* phase prioritizes the generation of core functions and progressively expands to dependent functions, producing functionally equivalent and architecture-independent

scalar migration code. **(3)** the *architecture-aware performance optimization* phase applies two levels of optimization: architecture-independent optimization logic from GFDG, such as algorithmic refinements and data layout adjustments, and architecture-specific optimization strategies for the target architecture, including register grouping and vector extension. **(4)** The *multi-stage testing* phase conducts comprehensive validation, including compilation tests, functional tests, and performance evaluations. If errors are detected, the results are fed back to trigger correction and re-optimization. Through this closed-loop process, FSCM ensures that the final migrated code is not only functionally correct but also efficient on the target architecture.

Preliminary experiments show that FSCM with GPT-5 achieves a success rate of 71% in OpenCV migration tasks, significantly outperforming the agent-model combinations.

## 7. Related Work

**Code Agents.** Early work such as GitHub Copilot (GitHub, 2025) and OpenAI Codex (OpenAI, 2023) demonstrated the potential of LLMs as coding assistants. Dong et al. (2024) proposed a framework where multiple LLM agents collaborate to tackle problems. SWE-Agent (Yang et al., 2024) introduced custom agent-computer interfaces for LLMs to interact autonomously. OpenHands (Wang et al., 2025c) provides a platform for developing AI code agents. Commercial AI-native IDEs such as JoyCode (JD, 2025) and TRAE (ByteDance, 2025a) have further advanced agentic coding capabilities in production environments. More recently, CodeAgent (Zhang et al., 2024) enhances repository-level code generation through tool-integrated agent systems, and LocAgent (Chen et al., 2025) improves code localization via graph-guided reasoning. While these agents excel in general coding tasks, none of them focuses on the challenge of library-scale SIMD migration.

**Code Migration.** Traditional code migration relies on manual or rule-based approaches (Feldman, 1990). Recent LLM-based methods have significantly improved the performance of code migration across programming languages (Eniser et al., 2024; Ibrahimzada et al., 2025). Pan et al. (2024) reveal that LLMs produce correct translations only 2.1%–47.3% of the time. To enhance migration accuracy, LLMs leverage rules(Zhang et al., 2025a; Nitin et al., 2025), symbolic methods, program analysis (Ibrahimzada et al., 2025; Shetty et al., 2024), or LLM-based methods (Wang et al., 2025a) to explore high-level code structure and semantics. For SIMD-specific migration, IntrinTrans (Han et al., 2025b) presents a multi-agent system to translate intrinsics for RISC-V Vector. LLM-Vectorizer (Taneja et al., 2025) uses finite-state-machine multi-agent approaches for vectorization. VecTrans (Zheng et al., 2025) enhances auto-vectorization through LLM-guided code refactoring.

Skeleton-Guided-Translation (Zhang et al., 2025b) provides fine-grained evaluation for repository-level translation. QM-LibBench differs by focusing on library-scale, cross-architecture SIMD migration with real-world build systems and performance evaluation.

**Code Generation Benchmarks.** Competitive programming benchmarks are represented by LiveCodeBench (Jain et al., 2024), OJBench (Wang et al., 2025d), and EffiBench (Huang et al., 2024), which evaluate models on dynamic problem solving and efficiency. General-purpose code generation benchmarks span various granularities, from function level (Chen, 2021) to class level (Du et al., 2023) and library level (Jimenez et al., 2023; He et al., 2025a). For software engineering tasks, various benchmarks have been proposed to evaluate LLM agents' capability to handle compositional function calls (Zhuo et al., 2024), debugging (Tian et al., 2024), test generation (Wang et al., 2025b; Mündler et al., 2024), and retrieval-augmented generation (Wang et al., 2025e). For tasks with larger scales, benchmarks for long horizon or code evolution are further explored, including SWE-Bench Pro (Deng et al., 2025) for long-horizon tasks, EvolveCodeBench (Li et al., 2024) for assessing code evolution in repositories, and AgencyBench (Li et al., 2026) for long-context agentic tasks. For tasks with larger scales, benchmarks for long horizon or code evolution are further explored, including SWE-Bench (Deng et al., 2025) for long-horizon tasks. For high-performance code generation, extensive work has been done on GPU kernel generation (Ouyang et al., 2025), and SIMD-intrinsic code generation (He et al., 2025b). VecIntrinBench (Han et al., 2025a) addresses cross-architecture intrinsic migration to RISC-V Vector at the function level. While these benchmarks have advanced LLM evaluation, none assess the unique challenges of *library-scale migration* involving complex build systems, cross-architecture semantics, and performance optimization.

## 8. Conclusion

We introduce QM-LibBench, the first comprehensive benchmark for evaluating LLM-based code agents on library-level cross-architecture SIMD migration. QM-LibBench comprises 85 performance-critical kernels with 12 state-of-the-art agent-LLM combinations. Extensive experiments reveal a peak correctness of only 20.88%, highlighting that current agents resort to superficial pattern matching rather than architectural reasoning. These findings expose critical limitations in handling library-level dependencies and architecture-aware optimization.

## 9. Limitations

Despite the broad coverage of QM-LibBench, several limitations remain.

**Benchmark longevity.** LLM-based agents are advancing rapidly, and future systems may gradually saturate parts of the benchmark. Although current state-of-the-art agents still achieve relatively low correctness and performance on QM-LibBench, the current benchmark scale may eventually become insufficient for long-term evaluation.

**Reproducibility of commercial agents.** Our evaluation includes commercial IDE-based agents such as TRAE and JoyCode, whose internal implementations and orchestration strategies are not publicly accessible. Although these systems are widely adopted and competitive on SWE-Bench, their closed-source nature limits full reproducibility and fine-grained analysis.

**Coverage of migration paths.** QM-LibBench currently focuses on representative migration directions, including x86→ARM, ARM→RISC-V, and ARM→LoongArch, covering both mainstream and emerging SIMD paradigms. However, other practically relevant scenarios, such as GPU-oriented migration, are not yet included.

**Dataset construction bias.** QM-LibBench is built through a curated selection process prioritizing actively maintained and performance-critical operators from widely used libraries. While this improves practical relevance and evaluation reliability, it may bias the benchmark toward operators with mature testing infrastructure and clearer SIMD optimization patterns.

## Acknowledgments

We thank all reviewers for their valuable feedback. This work is partially supported by the NSF of China (under Grant 92364202), and Major Program of ISCAS (Grant No. ISCAS-ZD-202402).

## Impact Statement

QM-LibBench introduces the first comprehensive benchmark for evaluating AI agents on library-level SIMD code migration across diverse hardware architectures. By automating the porting of compute-intensive libraries to emerging architectures, this work enables faster maturity of heterogeneous architecture ecosystem. On one hand, it may democratize access to high-performance computing by lowering technical barriers for adopting new architectures. On the other hand, automated code migration capabilities could be misused if applied maliciously. We believe the positive impact of enabling broader hardware accessibility far outweighs these risks.

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

# Appendix

## A. Libraries in QM-LibBench

- **OpenCV.** As the de facto standard for computer vision, OpenCV features a massive codebase with a complex Hardware Abstraction Layer (HAL). It tests the agent's ability to navigate high-level abstractions and diverse algorithmic logic.

- **ncnn.** Ncnn is a high-performance neural network inference framework for mobile platforms. It is characterized by aggressive, assembly-like intrinsic usage and intricate memory access patterns, providing a rigorous test for performance optimization.

- **libjpeg.** Libjpeg is a fundamental multimedia library representing compute-bound image compression logic. It requires the agent to handle arithmetic-heavy loops with strict precision requirements.

## B. Selection of Agents and Settings

To ensure representativeness in our evaluation of LLM-based code agents, we ground our selection in SWE-Bench (Jimenez et al., 2023), which is a widely-recognized benchmark for large-scale code modification, particularly in tasks requiring cross-file reasoning, dependency understanding, and iterative debugging.

Based on publicly reported SWE-Bench leaderboards and technical reports, we select 4 code agents that demonstrate strong empirical performance and substantial community adoption: TRAE (ByteDance, 2025a), SWE-Agent (Yang et al., 2024), JoyCode (JD, 2025), and OpenHands (Wang et al., 2025c). These agents represent leading approaches to autonomous code editing over real-world libraries. By anchoring agent selection to SWE-Bench performance rather than ad hoc choices, we align our experimental protocol with established evaluation practices and ensure that the studied systems reflect the current frontier of library-level coding agents.

To ensure reproducibility, we document the exact configurations used for each agent framework in our experiments.

### B.1. SWE-Agent

We use the open-source version of SWE-Agent with its default execution workflow. The core inference configuration is:

- `temperature = 0.0`

- `top_p = 1.0`

- `convert_system_to_user = false`

- `retry = {retries: 20, min_wait: 10.0, max_wait: 120.0}`

- `delay = 0.0`

- `fallbacks = []`

- `choose_api_key_by_thread = true`

- `litellm_model_registry = null`

- `custom_tokenizer = null`

- `max_requeries = 3`

- `action_sampler = null`

- `type = default`

All other system behaviors follow the official SWE-Agent open-source implementation without modification.

## B.2. OpenHands

OpenHands experiments were conducted using the official Docker image (version **0.62.0**). We selected this version due to its stable tool invocation and execution environment management. Default agent planning and tool-use configurations provided by this release were retained.

## B.3. JoyCode

JoyCode experiments were performed using the official AI-native IDE client (version **v2.2.5**). The **context compression** feature was enabled to mitigate long-context accumulation during multi-file repository interaction. All other IDE and agent behaviors follow the default production configuration.

## B.4. TRAE

TRAE experiments were conducted using the official IDE-based environment provided by ByteDance. The built-in project navigation, persistent workspace, and toolchain integration features were used as provided, without additional customization.

# C. Cost Analysis

This section provides a comprehensive breakdown of the resource consumption of different agents during migration on QM-LibBench. While the main paper reports summarized cost trends, Tables 5 7 present the full per-library statistics for OpenCV, ncnn, and libjpeg under both successful and failed migration trajectories.

We measure cost along five orthogonal dimensions:

- **Time**: End-to-end wall-clock time (in seconds) required to complete one migration attempt.

- **Money**: Estimated API expenditure based on token usage and model pricing.

- **API Calls**: Total number of model invocation rounds.

- **Input Tokens**: Total prompt tokens consumed.

- **Output Tokens**: Total generated tokens.

Overall, cost is not strictly correlated with correctness. Some agents achieve marginal accuracy gains at the expense of 2–4× higher resource consumption, indicating that current agent frameworks lack mechanisms for cost-aware planning and trajectory pruning. Improving migration practicality therefore requires not only better reasoning but also efficiency-oriented agent control strategies.

*Table 5.* OpenCV Cost Comparison (SR@1)

| Costs | Successful Migration | | | | Failed Migration | | | |
|---|---|---|---|---|---|---|---|---|
| | Time | Money | API Calls | I/O Tokens | Time | Money | API Calls | I/O Tokens |
| *SWE-Agent (Yang et al., 2024)* | | | | | | | | |
| claude-sonnet-4-5 | / | / | / | / | 442 | 0.73 | 17 | 256K/3K |
| gemini-3-pro-preview | / | / | / | / | 1195 | 2.09 | 95 | 5200K/- |
| GPT-5 | 649 | 1.32 | 37 | 1512K/3K | 863 | 1.58 | 41 | 1552K/2K |
| *JoyCode (JD, 2025)* | | | | | | | | |
| claude-sonnet-4-5 | 170 | 2.41 | 19 | 541K/7K | 244 | 4.01 | 22 | 1133K/10K |
| gemini-3-pro-preview | 253 | 2.06 | 16 | 979K/9K | 360 | 2.85 | 19 | 1287K/23K |
| GPT-5 | 424 | 1.72 | 18 | 1201K/21K | 608 | 1.56 | 16 | 1061K/24K |
| *Openhands (Wang et al., 2025c)* | | | | | | | | |
| qwen3-coder-480b | 1109 | / | 100 | 2522K/19K | 1627 | / | 162 | 4201K/29K |
| gemini-3-pro-preview | 944 | 1.30 | 57 | 1200K/17K | 817 | 2.01 | 67 | 1847K/28K |
| GPT-5 | 1075 | 1.94 | 80 | 1708K/286K | 1109 | 2.25 | 79 | 1951K/622K |

*Table 6.* ncnn Cost Comparison (SR@1)

| Costs | Successful Migration | | | | Failed Migration | | | |
|---|---|---|---|---|---|---|---|---|
| | Time | Money | API Calls | I/O Tokens | Time | Money | API Calls | I/O Tokens |
| *SWE-Agent (Yang et al., 2024)* | | | | | | | | |
| claude-sonnet-4-5 | 542 | 2.41 | 32 | 2197K/2K | 427 | 2.62 | 36 | 918K/3K |
| gemini-3-pro-preview | 571 | 2.95 | 65 | 2360K/- | 1095 | 1.74 | 73 | 174K/- |
| GPT-5 | 520 | 1.45 | 36 | 1550K/2K | 586 | 1.53 | 36 | 1715K/36K |
| *JoyCode (JD, 2025)* | | | | | | | | |
| claude-sonnet-4-5 | 210 | 2.07 | 16 | 633K/11K | 471 | 3.50 | 25 | 1012K/31K |
| gemini-3-pro-preview | 885 | 3.34 | 34 | 1175K/82K | 769 | 2.89 | 26 | 958K/68K |
| GPT-5 | 763 | 1.49 | 22 | 785K/50K | 1303 | 2.37 | 31 | 1178K/86K |
| *Openhands (Wang et al., 2025c)* | | | | | | | | |
| qwen3-coder-480b | 1074 | / | 83 | 2584K/17K | 1847 | / | 129 | 4440K/37K |
| gemini-3-pro-preview | 541 | 0.98 | 65 | 1027K/19K | 646 | 1.61 | 51 | 1329K/26K |
| GPT-5 | 1054 | 2.41 | 80 | 1894K/34K | 1386 | 2.60 | 87 | 2485K/45K |

*Table 7.* libjpeg Cost Comparison (SR@1)

| Costs | Successful Migration | | | | Failed Migration | | | |
|---|---|---|---|---|---|---|---|---|
| | Time | Money | API Calls | I/O Tokens | Time | Money | API Calls | I/O Tokens |
| *SWE-Agent (Yang et al., 2024)* | | | | | | | | |
| claude-sonnet-4-5 | / | / | / | / | 714 | 2.65 | 38 | 843K/2K |
| gemini-3-pro-preview | / | / | / | / | 1545 | 4.44 | 81 | 4437K/- |
| GPT-5 | / | / | / | / | 1081 | 2.02 | 53 | 2142K/1K |
| *JoyCode (JD, 2025)* | | | | | | | | |
| claude-sonnet-4-5 | / | / | / | / | 725 | 3.50 | 25 | 1012K/31K |
| gemini-3-pro-preview / | / | / | / | / | 831 | 2.89 | 26 | 958K/68K |
| GPT-5 | / | / | / | / | 1557 | 2.37 | 31 | 1178K/86K |
| *Openhands (Wang et al., 2025c)* | | | | | | | | |
| qwen3-coder-480b | 542 | / | 78 | 1512K/7K | 2721 | / | 197 | 6536K/60K |
| gemini-3-pro-preview | 832 | 3.12 | 101 | 2982K/23K | 619 | 1.90 | 61 | 1765K/197K |
| GPT-5 | 3163 | 1.78 | 83 | 1950K/43K | 1579 | 1.72 | 84 | 1894K/44K |

## D. Failure Mode Analysis.

We analyzed the distribution of failure error types across different agents on QM-LibBench, revealing distinct behavioral patterns in handling cross-architecture SIMD migration tasks.

**OpenHands** demonstrates a balanced error profile with notable strengths in avoiding syntax and dependency-related failures ($\leq 2.1\%$ each), benefiting from its iterative debugging approach. However, it exhibits the highest rate of functional correctness errors (78.5%), suggesting that while its debugging mechanism resolves surface-level issues, it struggles with deeper algorithmic semantics and architecture-specific optimizations.

**JoyCode** shows the highest proportion of compilation errors (77.2%), consistent with its IDE-based workflow that emphasizes early syntax and build validation. This agent maintains moderate functional correctness errors (12.2%), indicating better semantic preservation than OpenHands but still significant room for improvement in algorithmic translation.

**SWE-Agent** displays the most distributed error pattern, with high rates in undefined identifiers (26.0%), parameter mismatches (14.2%), and dependency/build errors (13.8%). This reflects its issue-level workflow from SWE-Bench, which lacks proper adaptation to library-scale migration contexts and architecture-specific constraints.

**TRAE** exhibits moderate performance across categories, with relatively high compilation errors (68.1%) but balanced distribution in other failure types (3.5%-19.7%). Its integrated IDE environment helps mitigate some semantic errors but still faces challenges with architectural translation correctness.

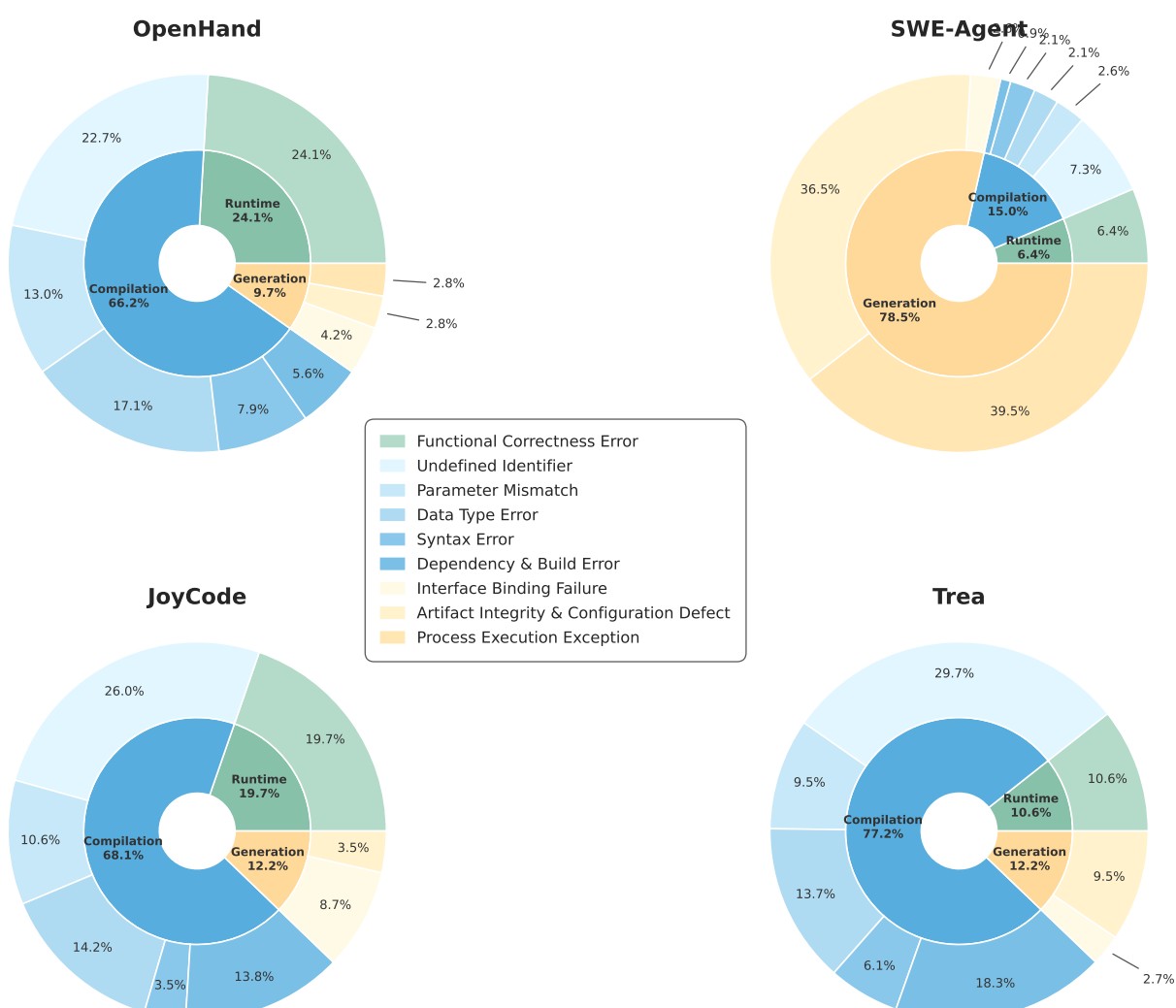

*Figure 6.* Summary of Failure Error Types.

*Table 8.* Cost and Performance Comparison of Different Agents on OpenCV.

| Agent | SR@1 | SR@3 | Time | Money | API Calls | Input Tokens | Output Tokens |
|---|---|---|---|---|---|---|---|
| SWE-Agent | 9.52 | 23.8 | 585 | 1.38 | 36 | 1531K | 2K |
| JoyCode | 14.29 | 28.57 | 593 | 1.6 | 20 | 993K | 36K |
| OpenHands | 28.57 | 38.1 | 1764 | 2.04 | 81 | 1850K | 121K |
| ClaudeCode | 28.57 | 52.38 | 816 | 3.99 | 56 | 383K | 31K |
| OpenCode | 33.33 | 52.38 | 429 | 1.75 | 24 | 648K | 21K |

# E. FSCM.

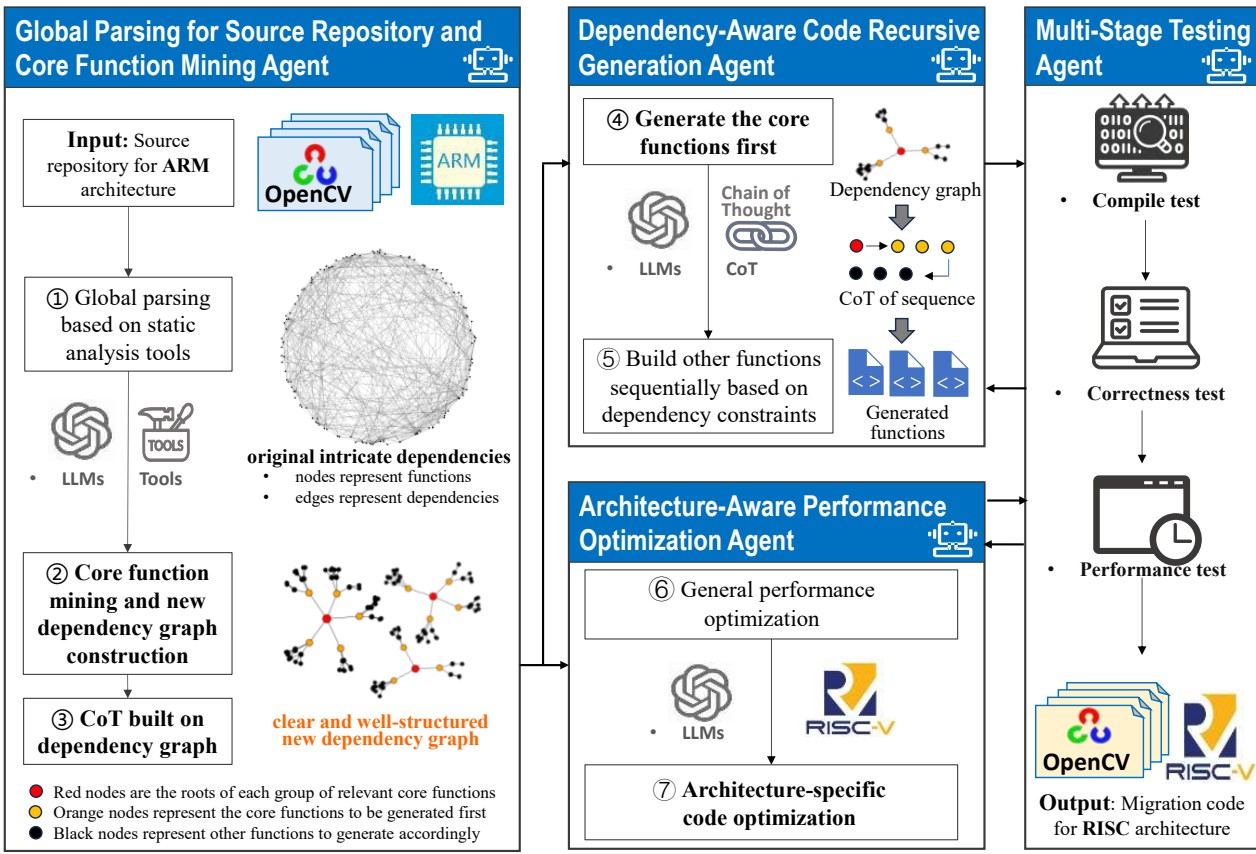

*Figure 7.* Overview of FSCM. It guides LLM agents to first construct a global functional dependency graph and identify core functions, and then recursively generate related functions, significantly enhancing the efficiency and performance of complex migration tasks.

# F. Additional Experiment.

We further evaluated two additional agents, ClaudeCode and OpenCode, both powered by GPT-5, on OpenCV migration tasks. The results are shown in 8:

OpenCode demonstrates the strongest overall performance, achieving top correctness with lowest runtime and cost. This further supports our conclusion that agent design matters substantially—even with the same underlying LLM, different agents exhibit significant variation in both success rate and resource efficiency.

