# OpenReview forum: "QiMeng-LibBench: Benchmarking LLM Agents for Library-Scale Cross-Architecture Migration"
_ICML.cc/2026/Conference — ICML 2026 regular_

### Official Review · Reviewer_bJbF · 2026-03-11

**Soundness:** 2
**Presentation:** 2
**Significance:** 2
**Originality:** 2
**Overall Recommendation:** 2
**Confidence:** 4

**Summary:**

CLAM-Bench is the first benchmark designed to evaluate LLM agents on library-scale cross-architecture migration tasks. The benchmark reflects the growing importance of heterogeneous computing, where modern systems must support multiple CPU architectures with architecture-specific SIMD intrinsics, while also addressing the fundamental challenges of library-scale migration, such as large codebases, complex dependencies, and tightly coupled hardware-specific optimizations. It includes 85 performance-critical kernels mined from industrial libraries such as OpenCV, libjpeg, and NCNN, evaluating migrations across ARM, x86, RISC-V, and LoongArch. Experimental results show that current LLM agents, which largely rely on superficial pattern matching, achieve only 20.88% correctness on average, indicating significant limitations in handling library-scale migration and hardware-aware optimizations. However, the proposed FSCM method substantially improves migration correctness, reaching up to 71% correctness in OpenCV migration tasks.

**Compliance With Llm Reviewing Policy:**

Affirmed.

**Key Questions For Authors:**

1. How well does FSCM generalize beyond the OpenCV + GPT-5 setting (e.g., other libraries or weaker LLMs)?

2. Given that different agents are paired with different LLMs, how should the comparison results be interpreted with respect to agent design vs. model capability?

3. Could the authors provide more details on the experimental setup (e.g., compiler version, optimization flags, build environments) to facilitate reproducibility?

**Limitations:**

Yes

**Strengths And Weaknesses:**

Strengths

(+) The paper introduces the first library-scale benchmark for cross-architecture code migration.

(+) The paper provides a quantitative analysis of the limitations of current LLM agents in handling cross-architecture migration tasks.

Weaknesses

(-) Limited dataset coverage
The benchmark evaluates only three libraries and 85 kernels, focusing mainly on isolated operators rather than full library-level systems. Moreover, the dataset is heavily biased toward C/C++ image processing and deep learning workloads (OpenCV, ncnn, libjpeg), while other important domains such as HPC, networking, and multimedia systems are not represented. Furthermore, the evaluated architecture transitions are limited, with some common scenarios such as x86-to-RISC-V migration not considered. This raises concerns about whether the benchmark fully reflects the diversity of real-world library-scale migration scenarios.

(-) Limited evaluation of the proposed FSCM framework
The proposed FSCM framework is primarily validated on OpenCV with GPT-5, leaving its generalizability unclear. It is not demonstrated whether FSCM maintains similar performance across other libraries or when paired with weaker LLMs. Moreover, the paper does not report any cost analysis, despite the recursive code generation and multi-stage testing loop, which may incur significant API token usage and runtime overhead. Therefore, the practical deployability and scalability of FSCM remain uncertain.

(-) Metric limitations
Correctness is measured using pass@k, which verifies test-case success but does not necessarily guarantee full functional equivalence, particularly under unspecified edge cases. The performance metric (Speedup Ratio) directly compares runtimes across architectures, which may conflate code quality with hardware disparities, since differences in microarchitectural characteristics, parallel execution capabilities, and compiler optimizations are not explicitly controlled for. Furthermore, the fast_p metric is coarse-grained, treating even minor slowdowns as failures due to its binary threshold design.

(-) Confounded agent–LLM attribution
Different agents are paired with different underlying LLMs, making it difficult to disentangle improvements due to agent design from those attributable to base model capability. This confounding factor weakens the causal interpretation of the comparative results.

(-) Insufficient experimental specification
Critical implementation details such as compiler version, optimization flags, and build environments are not clearly specified. This lack of transparency may hinder faithful reproduction of the reported results, particularly across different hardware setups.

(-) Lack of process-level evaluation
The evaluation focuses primarily on final correctness and runtime performance, without systematically analyzing migration efficiency or code quality. Although the paper reports some cost statistics, it does not provide explicit metrics for evaluating generation efficiency. Important process-level factors such as API call frequency and code maintainability are therefore not systematically assessed.

---

> ### Author Rebuttal · Authors · 2026-03-31
>
> We thank the reviewer for the feedback. We address each concern below and will incorporate the suggested clarifications and additional analyses in the revised version.
> ## 1. Limited Evaluation of FSCM
>
> We present FSCM as a validated design direction, not an exhaustive system. CLAM-Bench shows that existing agents rely on superficial pattern matching, failing at library-scale SIMD migration. FSCM shows that explicit dependency graph reconstruction can address this gap.
>
> ### 1.1 Generalizability of FSCM
>
> We selected OpenCV + GPT-5 because OpenCV is the most complex and diverse library (1,341K LoC) among three, and GPT-5 is the only model available across by 3 baseline agents, ensuring fair comparison.
> Additional experiment supports the generalizability.
> Setting|Configuration|Pass@1
> -|-|-
> Different Model|OpenCV+Claude-Sonnet-4.5|78%(vs. GPT-5’s 71%)
> Different Library|ncnn(ARM NEON→RISC-V RVV)+GPT-5|76.19% (vs. OpenHands’ 41.18%)
>
> These results suggest FSCM generalizes across models and libraries.
> ### 1.2 Cost Analysis
> FSCM’s dependency graph reconstruction narrows the search space and controls overhead. Below is the cost comparison with baseline agents on OpenCV:
> Agent|Time|Money|API Calls|Input Tokens|Output Tokens
> -|-|-|-|-|-
> SWE-Agent|585|1.38|36|1531K|2K
> JoyCode|593|1.6|20|993K|36K
> OpenHands|1764|2.04|81|1850K|121K
> FSCM|660|1.88|38|1438K|35K
> ## 2. Agent–LLM Confounding
>
> We agree that separating agent design from model capability is ideal, but not fully feasible in practice.
>
> SWE-Agent, JoyCode, and OpenHands all support GPT-5, Gemini-3-pro, and Claude-Sonnet-4.5. However, OpenHands+Claude failed to produce executable code in preliminary tests, so we used Qwen3-Coder-480B instead. For the closed-source TRAE IDE, model choice is restricted by its ecosystem, preventing orthogonal control.
>
> Despite these constraints, we can still isolate agent design effects where possible. Among GPT-5-selected (SWE-Agent, JoyCode, OpenHands), OpenHands achieves the highest Pass@1 (37.65%), suggesting agent design matters beyond model capability.
>
> We will note this confounding issue in the Limitations section.
> ## 3. Reproducibility
> Due to space constraints, please refer to Reviewer mPDm's response 2.
> ## 4. Benchmark Coverage and Diversity
> We agree that border benchmark coverage is valuable and plan to expand it in follow-up work. We clarify the current scope below.
> ### 4.1 Not Isolated Operators
> As shown in Table 1, each operator migration requires navigating 717K LoC, across 10 files and 51 functions, editing 967 LoC and 3 files on average, far beyond typical SWE-bench tasks (~1.7 files, ~33 lines). Migrating a single operator thus demands understanding the entire library, and is a genuine library-scale task.
> ### 4.2 Domain Coverage
> While OpenCV, ncnn, and libjpeg are from image processing and DL, their operators reflect patterns common across domains: dense linear algebra (convolution), memory-bound transforms (resize), bit-level precision (Huffman), and irregular control flow (Canny). These are relevant to HPC, multimedia, and networking.
> ### 4.3 Architecture Transitions
> Our benchmark includes 3 diverse scenarios: x86→ARM, ARM→RISC-V, and ARM→LoongArch. We will add x86→RISC-V in the future work.
> ### 4.4 Benchmark Scale
> SIMD migration benchmarks are usually smaller than high-level code generation ones, but each task is more complex because correctness depends on hardware-specific properties such as intrinsic mapping and vector semantics. Our contribution lies in per-task complexity and library-scale context, not raw count. Our scale is consistent with prior SIMD work such as VecIntrinBench (50), PolyBench (30), and SIMDBench (136).
> ## 5. Metric Limitations
> ### 5.1 Pass@k
> We acknowledge that pass@k does not guarantee full functional equivalence. For OpenCV and ncnn, we use official test suites validated by the community. For libjpeg, we add edge cases (e.g., boundary dimensions, extreme quantization tables) to increase coverage.
>
> More broadly, test-based correctness is a common limitation of code generation benchmarks, not unique to CLAM-Bench. We will note this in the Limitations section.
> ### 5.2 Speedup Ratio(SR)
> SR compares code on the same target architecture, using the library's existing target-architecture implementation as the baseline. Hardware disparity is therefore not a confounding factor.
>
> ### 5.3 Fast_p
>
> We use fast_p following KernelBench. Though binary, it practically reflects a kernel is faster or not. Continuous metrics such as geometric mean can be skewed by outliers, while fast_p is cleaner and more robust. Varying p also yields a fine-grained performance profile. We therefore consider it suitable for evaluating deployable high-performance code.
>
>
> ## 6. Lack of Evaluation
>
> Section 5.3 and Appendix C report time, cost, API calls, and I/O token consumption for each agent-LLM pair, with analysis of their relation to migration success/failure. Systematic code quality analysis is considered for future work.

---

> > ### Author Rebuttal · Reviewer_bJbF · 2026-04-03
> >
> > Fully resolved, but the score remains because the work has many limitations confirmed by the rebuttal.

---

> > > ### Author Response · Authors · 2026-04-07
> > >
> > > Thank you for acknowledging that all your concerns have been fully resolved. We are glad that our reply have addressed the methodological issues.
> > >
> > > We would like to respectfully clarify that our rebuttal did not introduce new weaknesses, but rather contextualized these limitations and explained why they do not undermine the validity or contribution of the work. Therefore, maintaining the original score solely on the basis that these limitations were confirmed, does not reflect an updated assessment of the paper after rebuttal. We will further elaborate on all possible issues below.
> > >
> > > ## Benchmark Coverage and Domain Diversity
> > >
> > > ### About Library
> > >
> > > We have already explained the library selection process to reviewer `oK8q`:
> > >
> > > > We began by mining GitHub for repositories containing SIMD headers (e.g., <riscv_vector.h>). We applied objective filters: Stars > 2k, active maintenance, and build system support. We also measured SIMD intrinsic density to ensure that candidate projects were genuinely SIMD-intensive, rather than using intrinsics only sporadically. This yielded candidates like OneDNN, MNN, XNNPACK, ncnn, OpenCV, and libjpeg. To ensure domain diversity, we selected one representative per domain: OpenCV (Vision), libjpeg (Image Codecs), and ncnn (Edge AI). We chose ncnn over similar AI libraries because its modular architecture effectively isolates SIMD kernels, making it ideal for operator migration benchmarking.
> > >
> > > Specifically, among the mentioned uncovered domains: HPC-style operators are already substantially represented in CLAM-Bench; networking libraries such as DPDK make very limited use of SIMD, primarily confined to performance-critical paths (e.g., prefetch and memory operations) rather than large-scale vectorized computation in operators, making them unsuitable as representative libraries for library-scale SIMD intrinsic migration benchmarking. Regarding multimedia, OpenCV and ncnn precisely the representative libraries. We believe that selecting well-established, widely adopted open-source libraries with high SIMD density is the primary consideration for constructing the first library-scale cross-architecture migration benchmark, rather than exhaustively covering every domain.
> > >
> > > ### About Scale
> > >
> > > Our set of 85 kernels is comparable in size to similar SIMD benchmarks such as VecIntrinBench (50 kernels), PolyBench (30 kernels), and SIMDBench (136 kernels). The key distinction lies in the complexity of each individual task rather than the sheer number of kernels.
> > >
> > > We clarify that CLAM-Bench, as the first benchmark for library-scale cross-architecture migration, is designed to evaluate whether LLM agents can handle the core challenges of this domain: library-level dependency navigation, ISA-specific intrinsic translation, and hardware-aware performance optimization. The concerns regarding domain coverage and benchmark scale are inherent scoping choices rather than fundamental flaws, and they do not undermine the benchmark's core purpose: assessing agents on library-scale tasks with realistic complexity, and the current benchmark is sufficient to reveal fundamental capability gaps in state-of-the-art agents and to validate improved approaches like FSCM. Reviewer `CbBq` raised similar concerns, and these concerns were fully resolved after discussion. We have also committed to adding a dedicated Limitations section in the camera-ready version.
> > >
> > > ## Metric Limitations
> > >
> > > ### Pass@k
> > > As we noted in the first phase of discussion, pass@k is a common limitation of code generation benchmarks, not unique to CLAM-Bench. We have taken extensive measures to make pass@k a more faithful proxy for correctness: for OpenCV and ncnn, we adopt their official test suites, which are widely used and validated by the community; for libjpeg, we augment the test suite with edge cases covering boundary dimensions, extreme quantization tables, and corner-case DCT coefficients. Furthermore, we validated our test harness on the original architecture where all tests passed, supporting its adequacy for verifying SIMD-specific correctness.
> > > Reviewer `mPDm` raised the similar concern, and these concerns were fully resolved after discussion.
> > >
> > > ### Speedup Ratio(SR)
> > >
> > > We clarified in the first phase that SR compares code within the same architecture.
> > >
> > > ### Fast_p
> > >
> > > We clarified in the first phase that Fast_p is a more suitable metric for CLAM-Bench than mean, which is susceptible to outliers, due to its greater robustness. Representative benchmark work such as KernelBench similarly adopts this metric. Furthermore, Fast_p should not be simplistically interpreted as a binary threshold, as its value dynamically changes with varying p.
> > >
> > > ## Agent–LLM Confounding
> > >
> > > We clarified in the first phase that this limitation arises from the Agent’s own ecosystem, which causes certain Agents to be unable to use some base models, rather than being a flaw of CLAM-Bench.

---

### Official Review · Reviewer_oK8q · 2026-03-12

**Soundness:** 3
**Presentation:** 3
**Significance:** 3
**Originality:** 3
**Overall Recommendation:** 4
**Confidence:** 3

**Summary:**

This paper introduces CLAM-Bench, a benchmark for evaluating code agents on library-scale cross-architecture SIMD migration. The benchmark contains 85 operator-level migration tasks drawn from OpenCV, ncnn, and libjpeg, and evaluates compilability, functional correctness, and performance across x86→ARM, ARM→RISC-V, and ARM→LoongArch settings. The paper evaluates 12 agent–LLM combinations and finds that current agents perform poorly overall, especially once both correctness and speedup are required. To support the claim that the benchmark captures a real unsolved challenge, the paper also presents FSCM, a specialized multi-agent framework, and reports substantially higher correctness on OpenCV.

**Compliance With Llm Reviewing Policy:**

Affirmed.

**Final Justification:**

I thank again for the authors additional response.

The authors clarified the details regarding benchmark curation pipeline.

In the additional response, the authors clarified that the low pass rate is not coming from some common dependencies that are hard to implement, but from the failure modes analyzed by the authors in the paper.

The authors showed additional results regarding the generalizability of FSCM. The authors also provided results with more baselines, showing that even the state-of-the-art coding agent is achieving low pass rate.

I think the authors addressed my concerns and hereby maintain the positive score.

**Key Questions For Authors:**

1. The paper should clarify the benchmark curation pipeline much more concretely. How exactly were the three libraries chosen, how were operators ranked or prioritized, and how much of this process depended on manual judgment?


2. The correctness numbers are strikingly low for all evaluated agents. How much of the difficulty comes from migrating the operator logic itself versus needing to implement or repair surrounding dependent APIs or build configurations?

3. How independent are individual benchmark tasks? If solving one task requires nontrivial edits to shared infrastructure or neighboring components, then pass rates may partly reflect cross-task coupling rather than per-operator migration difficulty.

4. The paper evaluates performance after migration, but what should readers regard as the realistic target? Is there an expert-written target implementation or some oracle reference that indicates the achievable speedup on the destination architecture?

5. Can the authors provide broader evidence, beyond correctness on one benchmark slice, that FSCM improves the migration outcomes in a way that supports the benchmark’s intended claims?

**Limitations:**

The paper has impact statement and limitation discussion.

**Strengths And Weaknesses:**

Strengths:

1. The paper tackles a realistic and timely problem. Cross-architecture migration of optimized library code is much closer to real systems work than the function-level or single-file settings used in many existing code benchmarks. I also think the benchmark direction is novel: combining library context, architecture-specific SIMD migration, and performance-sensitive evaluation is a meaningful step beyond prior coding benchmarks.

2.  The paper includes both correctness and performance evaluation for the tasks. It is relevant given that these SIMD libraries and tasks are often optimized for performance.

3. The paper does not only present a benchmark, but also provides a prototype solution FSCM to probe whether stronger task decomposition or coordination can make additional improvements on the problem.


Weaknesses:

1. It is still unclear how representative the libraries and tasks are of the broader space of library-scale migration problems. The paper says libraries and operators are prioritized based on domain importance, frequency of invocation, and optimization criticality, but the actual selection procedure remains underexplained. I was not sure how much manual judgment was involved. The operator filtering and prioritization process is not described concretely enough. The paper says operators that are frequently invoked in workloads are prioritized, but it does not really explain how this frequency or centrality is measured.

2. The reported correctness numbers are surprisingly low across almost all agents. Low performance can itself be an interesting result, but here the paper needs more explanation. It is not fully clear whether the difficulty comes from the intrinsic translation problem, from the need to implement or repair many dependent APIs around each operator, or other reasons. The paper would benefit from a clearer account of what an individual task actually requires beyond “migrate one operator,” and how much surrounding code must be created or modified for a task to succeed. Without that, it is hard to interpret whether the benchmark is exposing the intended challenge.


3. The benchmark evaluates performance improvement, but the paper does not sufficiently justify what level of speedup should be expected in these cross-architecture settings, or what the natural upper bound is. If the source implementation is already highly architecture-specific, then “performance after migration” is a hard target to interpret without a stronger baseline.

4. I also think the agent/model selection could be discussed more carefully. Basing agent choice on SWE-Bench popularity is understandable, but it does not always guarantee these are the most informative systems for this setting. I wondered how newer code agents for repository-scale editing like Claude Code or Open Code would perform here.

---

> ### Author Rebuttal · Authors · 2026-03-31
>
> We sincerely thank the reviewer for the thoughtful and constructive feedback. We are encouraged that the reviewer finds the problem setting realistic, the benchmark direction novel, and the inclusion of both correctness and performance meaningful. Below we address each concern in detail.
>
> ## 1. Benchmark Pipeline
>
> Given the vast and complex nature of open-source SIMD libraries, we adopted the approach combining automated filtering with principled manual judgment.
>
> ### 1.1 Library Selection
>
> We began by mining GitHub for repositories containing SIMD headers (e.g., <riscv_vector.h>). We applied objective filters: Stars > 2k, active maintenance, and build system support. We also measured SIMD intrinsic density to ensure that candidate projects were genuinely SIMD-intensive, rather than using intrinsics only sporadically. This yielded candidates like OneDNN, MNN, XNNPACK, ncnn, OpenCV, and libjpeg. To ensure domain diversity, we selected one representative per domain: OpenCV (Vision), libjpeg (Image Codecs), and ncnn (Edge AI). We chose ncnn over similar AI libraries because its modular architecture effectively isolates SIMD kernels, making it ideal for operator migration benchmarking.
>
> ### 1.2 Operator Selection
>
> First, we manually filtered out historical legacy and dead code to ensure all remaining candidates were active and relevant. Second, we profiled typical workloads to identify high-frequency operators and selected these "hotspots" as our benchmark targets. For example, for ncnn, we run all end-to-end models in the library, count the invocation frequency of each operator, and rank them accordingly.
>
> ## 2. Understanding the Low Correctness Results
>
> The low correctness numbers reflect the dual challenges of intrinsic translation and library-level dependency maintenance, both inherent to this problem setting. To clarify the sources of difficulty, we regrouped the failure modes from our analysis (Section 5.4, Figure 4) as follows:
> - Intrinsic Migration Errors (62.4%): Comprising Functional Correctness Errors (15.0%), Undefined Identifiers (21.7%), Parameter Mismatches (8.9%), Data Type Errors (11.8%), and Syntax Errors (4.9%).
> - Library-Scale Dependency Errors (27.5%): Comprising Dependency & Build Errors (10.0%), Interface Binding Failures (4.6%), and Artifact Integrity & Configuration Defects (12.9%).
> - Agent Execution Errors (10.1%): Timeouts or API limits exceeded.
> This breakdown confirms that the majority of failures stem from the migration problem itself, validating that CLAM-Bench primarily exposes the intended challenge rather than merely testing agents' ability to navigate build systems.
>
> ## 3. Task Independence
>
> We ensure full task independence. Each benchmark task is executed in isolation: the library codebase is reset to its original state before every task. Tasks do not share modifications, and there are no inter-task dependencies.
>
> ## 4. Interpretation of Performance
>
> We clarify that the reported speedup is measured against the existing SIMD implementation on the target architecture within the original library, rather than the source-architecture version. These expert-written implementations provide a practical and meaningful reference point. However, they should not be considered a strict upper bound on performance, as many operators continue to be refined and optimized over time. Accordingly, we treat them as a meaningful baseline rather than an absolute ceiling. We will make this definition explicit in the evaluation section.
>
> ## 5. About FSCM
> Due to space constraints, please refer to Reviewer bJbF’s Response 1 for the generalization experiments of FSCM on other libraries and models, and to Reviewer mPDm’s Response 5 for the ablation study of FSCM at different stages.
>
> ## 6. Task Complexity
> As shown in Table 1, each operator migration requires navigating an average of 717K lines of code, spanning 10 files and 51 functions. In practice, migrating a single operator demands understanding the broader library codebase and, on average, requires generating 967 lines of code and modifying 3 files to complete the task successfully.
>
> ## 7. Agent Selection
>
> We further evaluated two additional agents—ClaudeCode and OpenCode—both powered by GPT-5 on OpenCV tasks:
> Agent|Pass@1|Pass@3|Time|Money|API Calls|Input Tokens|Output Tokens
> -|-|-|-|-|-|-|-
> SWE-Agent|9.52|23.8|585|1.38|36|1531K|2K
> JoyCode|14.29|28.57|593|1.6|20|993K|36K
> OpenHands|28.57|38.1|1764|2.04|81|1850K|121K
> ClaudeCode|28.57|52.38|816|3.99|56|383K|31K
> OpenCode|33.33|52.38|429|1.75|24|648K|21K
>
> OpenCode demonstrates the strongest overall performance, achieving top correctness with lowest runtime and cost. This further supports our conclusion that agent design matters substantially—even with the same underlying LLM, different agents exhibit significant variation in both success rate and resource efficiency.

---

> > ### Author Rebuttal · Reviewer_oK8q · 2026-04-03
> >
> > I thank the authors for their detailed response. My concerns are mostly addressed.
> >
> > I have one clarification on my concern of task independence. I was not referring to during actual execution of the benchmark how it ensures the evaluation of one task does not affect the others. Rather, it was about the formulation of the tasks. How much of tasks may share common operator dependencies? One explanation of why existing baseline agents have such low pass rate is that a large number of tasks for depend on some common operators that are hard to implement. Therefore, pass rates may partly reflect cross-task coupling rather than per-operator migration difficulty.

---

> > > ### Author Response · Authors · 2026-04-03
> > >
> > > We thank the reviewer for careful reading and this clarification. We now understand the concern is about cross-task dependency coupling. We systematically analyzed the dependency structure of all operators, and conclude that there are no inter-operator dependencies. To be more specific, the intra-operator dependencies fall into two categories:
> > >
> > > **Operator-specific dependencies.**
> > > Some dependencies (e.g., `batchnorm.h` for `batchnorm.cpp`) are unique to a single operator. Since no other operator share this dependency, there is no cross-task coupling. Therefore, migrating the header file is part of the challenge in migrating the operator.
> > >
> > > **Shared dependencies across many operators.**
> > > Most operators in CLAM-Bench depend on common low-level utility functions and templates (e.g., `saturate_cast`, `border_interpolate`). These utilities are semantically simple (e.g., value clamping, boundary checking) and are shared across all operators. They provide basic auxiliary functionality rather than complex algorithmic logic.
> > > Critically, agents successfully migrate simple operators like `add` and `sub` that depend on the same utilities.  Experiment results show that migration of simple operators do not consistently fail, indicating that the algorithmic logic of these shared dependencies is not a significant contributor to the low pass@k.
> > >
> > > Thank you for the review!

---

### Official Review · Reviewer_CbBq · 2026-03-16

**Soundness:** 3
**Presentation:** 3
**Significance:** 3
**Originality:** 3
**Overall Recommendation:** 4
**Confidence:** 4

**Summary:**

This manuscript's principal theme concerns evaluating the capability of LLM-based agents to perform library-scale cross-architecture code migration. Overall, a critical problem examined by the article is the lack of realistic benchmarks for assessing automated migration of high-performance libraries across heterogeneous hardware architectures.
To address this gap, the authors introduce CLAM-Bench, a benchmark designed to evaluate LLM agents on real-world library-level migration tasks involving SIMD intrinsics. The benchmark includes 85 performance-critical kernels extracted from widely used libraries such as OpenCV, ncnn, and libjpeg. The tasks involve migrating implementations across different instruction set architectures (e.g., ARM→RISC-V, x86→ARM). The benchmark evaluates agent-generated code across three dimensions: compilability, functional correctness, and performance.
The authors further evaluate 12 agent–LLM combinations and demonstrate that current systems perform poorly, achieving only around 20.88% correctness. Motivated by these findings, they propose FSCM, a hardware-aware migration framework that significantly improves migration correctness.

**Compliance With Llm Reviewing Policy:**

Affirmed.

**Final Justification:**

I thank the authors for their detailed rebuttal. The clarification that $T_{original}$ is measured on the same target hardware, alongside the newly added `Scalar-O3` compiler baseline, adequately resolves my primary methodological concerns regarding the performance evaluation. Furthermore, the `vector-add` micro-benchmark effectively rules out the interference of an immature RISC-V toolchain, and the supplementary experiments of FSCM on `ncnn` provide necessary evidence of generalizability. Although the benchmark still exhibits noticeable limitations in overall scale and task diversity, the authors have supplied crucial experiments during the rebuttal phase to patch the methodological flaws present in the original submission, bringing the paper above the baseline acceptance threshold. Consequently, I am adjusting my score to a 4 (Weak Accept). The authors must ensure that the promised metric renaming (SR@k), the additional traditional baselines, and a dedicated limitations section are fully incorporated into the camera-ready version.

**Key Questions For Authors:**

What does T_original refer to in the speedup ratio definition? In a cross-architecture migration setting, comparing wall-clock times is only meaningful when both measurements are taken on the same target hardware. The paper describes T_original as "the execution time of the original code," but it is unclear whether this refers to the scalar baseline on the target platform or the SIMD implementation on the source platform. Could you clarify this? If both times are measured on the target hardware, this concern is resolved; if not, the interpretation of all performance results would need to be reconsidered.

How exactly is Pass@k computed? The paper states that Pass@k is used but does not specify whether it follows the unbiased estimator commonly used in code generation literature or a simpler empirical success rate. It would also help to know the total number of attempts per task. Some patterns in the correctness table — where pass@1 appears to be zero for a configuration while pass@3 is nonzero — seem difficult to reconcile with the standard unbiased estimator, suggesting an empirical rate may have been used instead. Clarifying the computation method would help readers properly interpret the correctness results.

Have rule-based translation tools been evaluated on CLAM-Bench? The introduction motivates the benchmark partly by noting that rule-based tools like neon2rvv produce suboptimal results, but no quantitative evidence from CLAM-Bench itself is provided. It would meaningfully strengthen the paper to include these as baselines. Similarly, a compiler auto-vectorization baseline (compiling de-intrinsified scalar code with aggressive optimization flags) would provide a useful lower bound. Without such reference points, it is hard to contextualize how well or poorly the agents actually perform.

Can you provide FSCM results beyond OpenCV? FSCM is presented as one of the paper's main contributions, but results are only reported for one library. Extending the evaluation to the other two libraries and providing ablation results for each stage of the pipeline would help readers assess whether the framework generalizes. In its current form, FSCM reads more as a preliminary case study than a fully validated contribution.

Could toolchain immaturity on the RISC-V platform confound the performance evaluation? The RISC-V hardware used is relatively new, and compiler support for the vector extension may be less mature than for ARM NEON. Is there evidence that the observed performance gaps on RISC-V targets reflect agent code quality rather than backend compiler limitations? Even a brief comparison of expert-written code against theoretical throughput on this platform would help disentangle these factors.

**Limitations:**

The authors include an Impact Statement that touches on positive societal impact (lowering barriers to new architecture adoption) and briefly acknowledges potential misuse risk. However, several methodological limitations are not discussed and would benefit from explicit treatment:

Benchmark longevity. Given the rapid pace of LLM improvement, the paper should discuss how quickly the benchmark might be saturated and whether its current scale provides sufficient headroom for future evaluation. A plan for versioning or expansion would be helpful.
Reproducibility of commercial agents. Two of the four agents evaluated are closed-source commercial IDEs whose internal behavior cannot be inspected or replicated. The paper should acknowledge this as a reproducibility constraint and discuss its implications for the reliability of those results.
Coverage of migration paths. The benchmark covers three migration directions but omits others that may be industrially relevant (e.g., x86 to RISC-V). Acknowledging this scope limitation and discussing whether the chosen paths are sufficiently representative would be appropriate.
Absence of a human expert reference. Without any data on how skilled engineers perform on the same tasks, it is difficult for readers to judge whether current agent performance represents a narrow or wide gap relative to human capability. Even a small-scale human study would improve interpretability.

I would encourage the authors to add a short, dedicated Limitations section addressing these points. Being transparent about these issues would strengthen rather than weaken the paper.

**Strengths And Weaknesses:**

Strengths
1. Novel Benchmark for Library-Scale Migration
The most significant contribution of this paper is the introduction of CLAM-Bench, which evaluates code generation systems on library-scale migration tasks. Unlike existing benchmarks that operate at the function or class level, CLAM-Bench captures realistic challenges such as build systems, cross-file dependencies, and architecture-specific intrinsics.

2. Realistic Industrial Workloads
The benchmark is constructed using widely deployed libraries (OpenCV, ncnn, and libjpeg), ensuring that the tasks represent real-world high-performance software engineering scenarios. This significantly improves ecological validity compared with synthetic benchmarks.

3. Multi-Dimensional Evaluation
The evaluation framework considers not only correctness but also compilability and runtime performance, which are essential for real-world deployment. The inclusion of the fast_p metric provides a meaningful measure of solutions that are both correct and performant.

4. Insightful Empirical Findings
The evaluation results provide useful insights into the limitations of current LLM agents. In particular, the paper highlights that many models rely on superficial pattern matching rather than deep understanding of architecture-specific semantics.



Weaknesses
1. Limited Benchmark Scale
Although the benchmark focuses on realistic libraries, the dataset contains only 85 kernels. Given the scale of modern software ecosystems, this size may be insufficient to fully represent the diversity of real-world migration challenges. A larger benchmark covering more libraries and operators would strengthen the conclusions.

2. Insufficient Analysis of Failure Modes
The paper reports low correctness rates for existing agents but provides limited analysis of why these failures occur. For instance, the paper briefly mentions issues such as pattern matching and dependency handling but does not present systematic error categorization or case studies that analyze common failure patterns.

3. Limited Baseline Diversity
The evaluation focuses primarily on LLM-based agents. However, the paper does not include strong non-LLM baselines, such as static analysis tools or rule-based migration systems beyond simple translators. Without these comparisons, it is difficult to determine whether LLM agents truly represent the best available approach.

4. FSCM Description Lacks Detail
The proposed FSCM framework appears promising, but its technical description is relatively brief. Key aspects such as agent coordination, optimization strategies, and hardware-aware reasoning mechanisms are not sufficiently detailed. This makes it difficult to fully understand or reproduce the approach.

5. Performance Evaluation Limitations
While the benchmark includes performance metrics, the paper does not thoroughly analyze the performance characteristics of generated code. For example, it would be beneficial to understand whether performance degradation stems from memory layout issues, vector width mismatches, or inefficient instruction mappings.

6. Lack of Generalization Experiments
The evaluation focuses only on the CLAM-Bench dataset. It remains unclear whether the findings generalize to other types of migration tasks, such as GPU kernels, compiler IR transformations, or cross-language code migration.

---

> ### Author Rebuttal · Authors · 2026-03-31
>
> We sincerely thank the reviewer for the constructive feedback and recognizing the novelty, realism, and multi-dimensional evaluation of CLAM-Bench. We address each concern below.
>
> ## 1. Clarification of Performance Metric (T_original)
>
> Both T_original and T_migrate are measured on the same target hardware. T_original refers to the execution time of the existing target-architecture implementation in the original library, not the source architecture or a scalar version. We will make this definition explicit.
>
> ## 2. Clarification of Computation Pass@k
>
> We use the empirical success rate rather than the standard unbiased estimator. For each task, we perform 3 independent generation attempts: pass@1 measures the proportion of tasks where the first attempt succeeds, while pass@3 measures the proportion where at least one of the 3 attempts succeeds.
>
> We use this metric because it reflects the e2e task completion, including exploration, iterative refinement, and self-correction, and thus better reflects an agent's practical usability in realistic development.
>
> We agree that “Pass@k” may be misleading. We will rename it “Success Rate @ turn k” (SR@k) and use this term consistently.
>
> ## 3. Response to Baselines
>
> ### 3.1 About neon2rvv
>
> Rule-based tools such as neon2rvv perform poorly on CLAM-Bench, and direct comparison would be unfair, so we do not include them as primary baselines. Applying neon2rvv required substantial manual effort, including modifying CMake files, adding platform detection, restructuring headers, and defining new macros, making it incomparable to fully autonomous agents. Even after manual adaptation, we observed that neon2rvv fails all cases due to fundamental mismatches across ISAs (fixed-width NEON vs. vector-length-agnostic RVV). For example, RVV vector types lack compile-time known sizes, causing compilation failures when used in structs/unions; NEON code often overloads functions based on vector length, and naive translation leads to errors. These issues are pervasive challenges in library-scale migration, and cannot be resolved by simple rule-based approaches.
>
> ### 3.2 About Compiler Auto-vectorization Results
>
> Thanks for the suggestion! The corresponding results have been committed to the anonymous repository ([Scalar-O3](https://anonymous.4open.science/r/figure-EEC5)).
>
> ## 4. About FSCM
>
> Due to space constraints, please refer to Reviewer bJbF’s Response 1 for the generalization experiments of FSCM on other libraries and models, and to Reviewer mPDm’s Response 5 for the ablation study of FSCM at different stages.
>
> ## 5. On RISC-V Toolchain Maturity
>
> To evaluate RVV compiler maturity, we ran a micro-benchmark of vector-add on the same hardware and compiler used in the main evaluation. It achieves 14.41 GFlops/s, 60% of theoretical peak (24 GFlops/s based on VLEN=256 bits, 2 vector instructions per cycle, and 1.5 GHz). This indicates that the compiler can generate near-optimal vector code for well-written kernels. Since all methods are evaluated under the same hardware and environment, differences reflect relative code quality.
>
> ## 6. Limited Benchmark Scale
>
> We agree that border benchmark and operator coverage are valuable and plan to expand it in follow-up work. Due to space constraints, please refer to Reviewer bJbF's response 4.
>
> ## 7. About Failure Mode Analysis
>
> We report the failure mode statistics and analyzed the most common modes in Section 5.4 and Figure 4. A more detailed classification and analysis is provided in Appendix D. A table summarizing the case-by-case failure study is in the anonymous repository.
>
> ## 8. FSCM Description
> Due to space constraints, complete FSCM implementation details and code are provided in the anonymous repository (link in the paper). In the revision, we will add more descriptions of FSCM in the appendix to improve reproducibility.
>
> ## 9. Performance Evaluation
>
> Manual inspection suggests 3 main sources of performance loss. The largest fraction comes from reduction/fusion-related issues (e.g., pooling, batchnorm, rmsnorm), followed by data movement / memory-layout issues in rearrangement-heavy kernels (e.g., medianfilter, morph, chroma_downsampling), and target-ISA adaptation issues such as vector-width mismatch, RVV tail handling, and register grouping. A smaller but notable portion comes from numerically delicate operators, where agents use conservative implementations to preserve semantics.
>
> ## 10. About Generalization
>
> Our work focuses on library-scale cross-architecture SIMD migration, and it is the first benchmark to do so. Our methods may be generalized, but other areas, such as GPU kernels and IR translation, are not in our scope. Future work may include generaliztion to other scenarios.
>
> ## 11. On Additional Limitations
>
> We agree that explicitly acknowledging these aspects will strengthen the paper. Thanks for the suggestion!

---

> > ### Author Rebuttal · Reviewer_CbBq · 2026-04-03
> >
> > I thank the authors for their detailed rebuttal, as the new micro-benchmarks and supplementary experiments effectively resolve some concerns regarding the performance metrics. I am raising my score to a Weak Accept (4), but this positive recommendation is strictly conditional upon the full integration of the promised metric renaming (SR@k), the additional traditional baselines, and a dedicated limitations discussion in the camera-ready version.

---

> > > ### Author Response · Authors · 2026-04-04
> > >
> > > Thank you very much for your positive feedback and for raising your score. We truly appreciate your constructive suggestions throughout the review process.
> > >
> > > We confirm that all the requested revisions will be fully incorporated in the camera-ready version:
> > >
> > >
> > > **Metric renaming (SR@k).** We will rename the metric to SR@k to more accurately reflect its definition and avoid potential confusion. A clear and formal definition will be included in the paper.
> > >
> > > **Traditional baselines.** We will include the traditional baselines in the experimental comparison, including rule-based translation tools (e.g., neon2rvv) along with detailed analysis. Additionally, we will add compiler auto-vectorization as a reference lower bound in the performance table (Table 3, [Scalar-O3](https://anonymous.4open.science/r/figure-EEC5)) to better contextualize the results.
> > >
> > > **Dedicated limitations discussion.** We will add a dedicated Limitations section that explicitly discusses important aspects, including benchmark longevity and potential saturation, reproducibility concerns for commercial agents, coverage of migration paths, and the scope of library selection.
> > >
> > > We sincerely thank you again for your careful reading and rigorous suggestions. We believe these revisions will significantly strengthen the paper.

---

### Official Review · Reviewer_mPDm · 2026-03-21

**Soundness:** 3
**Presentation:** 3
**Significance:** 3
**Originality:** 2
**Overall Recommendation:** 4
**Confidence:** 4

**Summary:**

CLAM-Bench is a benchmark for evaluating LLM agents on library-scale cross-architecture SIMD migration and on assembling performance-critical kernels from OpenCV, libjpeg, and ncnn. It measures compilability, correctness, and performance across ARM, RISC-V, and x86, revealing low pass rates and frequent pattern-matching failures. The paper analyzes 10+ agent/LLM combinations, identifies dominant failure modes, and proposes FSCM, a dependency-aware, multi-step framework that boosts OpenCV migration correctness to over 70% through core-function mining and hardware-aware optimization.

**Compliance With Llm Reviewing Policy:**

Affirmed.

**Final Justification:**

I have read the rebuttal. Thank you authors for your effort.

The authors have addressed my main concerns:

– They performed additional fine-tuning experiments that help separate training-data limitations from intrinsic SIMD semantic mismatch, this shows that much of the difficulty arises from required structural adaptations rather than simple token scarcity.

– They provided reproducibility details and I expect this will improve further in teh camera-ready version.

– They clarified fairness across agents by enforcing identical resource constraints while allowing tool-use differences to remain part of each agent’s design. I think this is pretty OK.

– They detailed the scalar ground truth and test suites, added edge cases and validation on the original architectures, which increases my confidence in the reported pass@k metrics that was originally concerned about.

– For FSCM, they confirmed apples-to-apples evaluation and added ablations tp sjow the impact of the dependency graph and multi-stage testing, substantially strengthening the methodological contribution.

They also provided a reasonable explanation of the observed performance cliff in terms of ISA-specific restructuring and hardware-aware execution semantics.

With these clarifications, my main concerns are adequately addressed, and I am raising my recommendation from weak reject to weak accept/borderline.

**Key Questions For Authors:**

Thank you for your timely work. I have a few questions, and I will consider improving my score.

Can you provide me with a breakdown of how much of the failure on harder targets is due to training data scarcity for those architectures versus the intrinsic difficulty of the SIMD semantic mismatches? Have you tried controlling for this?

Can you clarify what is required to reproduce the benchmark results end-to-end? I am thinking exact library commits, Docker images and toolchains, CPU governor and thermal controls, and how you handle run-to-run variance in performance testing.

Agents differ substantially in their workflows, and the paper notes that OpenHands benefits from command-line tool integration while others emphasize navigation and stability. How do you ensure that each agent is evaluated under comparable budgets, tool access, and stopping criteria, so the comparison reflects capability rather than scaffolding differences? If you can show careful normalization or ablations demonstrating robustness to budget and tool differences, it would improve the credibility of the head-to-head comparisons.

Can you clarify what exactly is considered the scalar ground truth for each library and operator, and how your test suites are strong enough to catch SIMD-specific edge cases like alignment, saturation, rounding, undefined behavior, and endian assumptions? Stronger evidence of test adequacy would increase my trust in the pass@k/3 numbers.

FSCM improves OpenCV correctness. Is this reported under the same pass@k setting and the same task set as the baseline agents, and which FSCM components contribute most (dependency graph vs. optimization vs. multi-stage testing)? For this reviewer, please include an apples-to-apples comparison plus component ablations, which would significantly strengthen the methodological contribution; without them, the improvement is harder to interpret.

**Limitations:**

Yes.

**Strengths And Weaknesses:**

+ The scope here is realistic and important. CLAM-Bench targets million-line, dependency-entangled libraries and requires consistent multi-file changes plus intrinsic rewrites. This is much closer to real migration work than function-only or scalar-only benchmarks. The large number of kernels is mined from widely used industrial libraries rather than synthetic tasks, which adds credibility.

+ Evaluation is multi-dimensional, which I like, and it covers compilability, functional correctness, and performance as orthogonal axes. This matters for SIMD migration, where "correct but slow" often fails in practice.

+ The benchmark does not only report low scores. I like that it analyzes failure modes and finds that agents often regress to superficial pattern matching, producing undefined identifiers, data type errors, parameter mismatches, and hallucinated APIs. This kind of diagnosis is useful.

+ The paper proposes and tests a remedy, not just a leaderboard. FSCM adds global dependency modeling, recursive dependency-aware generation, and architecture-aware optimization, and reports a meaningful improvement in correctness on OpenCV. I appreciate that the authors went beyond measuring and offered a concrete path forward.

- However, there is a risk of this becoming "yet another benchmark" if the community treats it as a score race. Although it is more realistic than many prior benchmarks, it still fundamentally reports pass@k and performance metrics, which can encourage chasing numbers rather than a deeper understanding.

- The benchmark primarily infers model behavior from observable artifacts rather than providing a systematic account of what models are actually doing internally. This is a fault with many papers, and this paper is no exception. The error categorization partly addresses this, but it remains an outcome-based evaluation used as a proxy for understanding. I would encourage the authors to think about how to push past this.

- Current agents perform poorly on the hardest requirements. Even strong agent/model combinations achieve limited correctness overall, and performance-aware success collapses quickly once you require no slowdown or speedup. I would like to see a deeper investigation into why this cliff exists and what it tells us about the limits of current approaches. I come from a systems background, and we really like to understand these nuances.

---

> ### Author Rebuttal · Authors · 2026-03-31
>
> Thank you for the thoughtful review and insightful suggestions. We appreciate your recognition of the paper’s strengths, including its realistic scope, multi-dimensional evaluation, and the constructive proposal of FSCM. We address your concerns below.
>
> ## 1. Failure Reason Analysis
>
> Distinguishing training data scarcity from intrinsic SIMD semantic mismatch is critical. To disentangle them, we conducted fine-tuning experiments.
> We constructed two datasets and fine-tuned Qwen2.5-7B:
>
> - Intrinsic Mapping (IM): 500 pairs of ARM NEON → RISC-V RVV intrinsic mappings.
> - Function-Level (FL): 150 simple functions synthesized from the IM data.
>
> To isolate library-level migration complexity, we evaluated both fine-tuned models on a function-level test set. The results show that IM fine-tuning achieves 40% pass@1, while FL fine-tuning achieves 58% pass@1. This suggests that knowledge alone is insufficient. The FL gain indicates that the main challenge lies in semantic mismatch—specifically, adapting to RVV's variable-length vector paradigm, which requires structural changes to loops, memory access patterns, and tail handling.
>
> ## 2. Response to Reproducibility Concerns
>
> To ensure reproducibility, we will provide the following in the camera-ready version:
>
> **Exact Library Versions**
> - OpenCV: v4.12.0 (commit 49486f6)
> - ncnn: 20250916 (commit c4193aa)
> - libjpeg: v3.1.80 (commit b5b660b)
>
> **Toolchains and Build Environment**
> Architecture|Compiler|Version
> -|-|-
> ARM|Arm GNU Toolchain|11.3.Rel1
> RISC-V|riscv64-unknown-linux-gnu-gcc|15.1.0
> LoongArch|LoongArch GNU toolchain|rc1.6
>
> **Performance Testing**
> We use 10-iteration warmup followed by 30 runs, and report mean execution time.
>
> **Compilation Flags**
> The detailed compilation flags is in the anonymous repository.
>
> ## 3. Fairness across Agents
>
> We clarify that noting OpenHands’ command-line integration does not mean other agents lack tools; all agents have tool access. Which tools are integrated and how they are used are part of each agent’s design. Enforcing identical tools would remove these defining differences. To ensure fair comparison, we instead used identical resource constraints for all agents: a 40-minute time limit and a 100 API call limit, as documented in Section 4 (Model and Constraints). These thresholds were set empirically: our experiments showed that tasks rarely succeed beyond these limits, and imposing these limits does not disproportionately harm any agent's performance. Together, these measures ensure that the reported differences reflect agent capability, including their tool-use strategies, rather than uneven resource allocation.
>
> ## 4. Response to Test Adequacy Concerns
>
> The scalar ground truth is the existing scalar C implementation validated through years of open-source use. For OpenCV and ncnn, we use official test suites. For libjpeg, we added edge cases including boundary dimensions, extreme quantization tables, and corner-case DCT coefficients. We also validated our test harness on the original architecture where all tests passed, supporting adequacy for SIMD-specific correctness. We acknowledge that no test suite can cover all possible edge cases. We have done our best to maximize coverage and will explicitly note this limitation in the paper.
>
> ## 5. Response to FSCM
>
> FSCM and baselines are evaluated on the same OpenCV tasks under identical task settings and pass@1. For ablation, we evaluated the following contents on OpenCV:
> Method|pass@1
> -|-
> w/o Dependency Graph|38.10%
> w/o Multi-stage Testing|47.62%
> FSCM|71%
>
> The reconstructed dependency graph is the most critical component. It breaks surface-level pattern matching by decoupling semantics from architecture-specific details and preserving only invariant anchors, thereby guiding the agent toward genuine deep functional semantics.  Multi-stage testing also matters by catching SIMD-specific errors that the dependency graph alone cannot resolve.
>
> ## 6. Performance Cliff
>
> Thank the reviewer for highlighting the performance cliff. Our analysis suggests that this drop is not random, but is concentrated on kernels whose high-performance implementations require target-ISA-specific restructuring rather than simple intrinsic translation. In these cases, agents often overfit source-architecture optimization patterns and fail to adopt execution strategies better suited to the destination architecture (e.g., RVV’s vector-length-agnostic execution, tail handling, and register grouping).
>
> This reveals a core limitation of current approaches: while they are often capable of surface-level functional translation, they lack the ability to reason about hardware-aware execution semantics and performance-critical transformations. As a result, the generated code may be functionally correct, yet still falls short of expert-written implementations, leading to the observed performance cliff. We will incorporate this analysis in the revised version.

---

> > ### Author Rebuttal · Reviewer_mPDm · 2026-04-05
> >
> > I have read the rebuttal. Thank you authors for your effort.
> >
> > The authors have addressed my main concerns:
> >
> > – They performed additional fine-tuning experiments that help separate training-data limitations from intrinsic SIMD semantic mismatch, this shows that much of the difficulty arises from required structural adaptations rather than simple token scarcity.
> >
> > – They provided reproducibility details and I expect this will improve further in teh camera-ready version.
> >
> > – They clarified fairness across agents by enforcing identical resource constraints while allowing tool-use differences to remain part of each agent’s design. I think this is pretty OK.
> >
> > – They detailed the scalar ground truth and test suites, added edge cases and validation on the original architectures, which increases my confidence in the reported pass@k metrics that was originally concerned about.
> >
> > – For FSCM, they confirmed apples-to-apples evaluation and added ablations tp sjow the impact of the dependency graph and multi-stage testing, substantially strengthening the methodological contribution.
> >
> > They also provided a reasonable explanation of the observed performance cliff in terms of ISA-specific restructuring and hardware-aware execution semantics. With these clarifications, my main concerns are adequately addressed, and I am raising my recommendation from weak reject to weak accept/borderline.

---

> > > ### Author Response · Authors · 2026-04-07
> > >
> > > We are glad that our clarifications have addressed your concerns. We truly appreciate your encouragement in reviewing our work.
> > >
> > > Thank you again for your positive response and valuable suggestions!

---

### Decision · Program_Chairs · 2026-04-30

**Decision:**

Accept (regular)

**Comment:**

This paper introduces CLAM-Bench, a benchmark for evaluating LLM agents on library-scale cross-architecture SIMD migration, and proposes FSCM, a multi-agent framework for improving performance on this task. Reviewers agree that the problem is important and the benchmark is novel in targeting realistic, library-scale migration with multi-dimensional evaluation. Three out of four reviewers gave 4. One reviewer maintains a reject (score 2), primarily due to concerns about limited scale/coverage, metric limitations, and perceived weaknesses in experimental design and generality. While these concerns are valid, any benchmark is necessarily limited in scope, and these limitations are largely acknowledged by the authors and partially mitigated through additional experiments and clarifications. One weakness is that the paper could provide a deeper and more systematic analysis of why existing agents fail, beyond reporting aggregate outcomes and high-level failure modes.

Overall, the paper makes a timely and valuable contribution by establishing a realistic benchmark for library-scale cross-architecture migration and providing empirical insights into the limitations of current LLM agents. It is likely to be useful for the community and to stimulate further research in this space.